# Policy Compatible Skill Incremental Learning via Lazy Learning Interface

**Daehee Lee**♠ **Dongsu Lee**◇ **TaeYoon Kwack**♠ **Wonje Choi**♠ **Honguk Woo**♠ *
♠Sungkyunkwan University    ◇University of Texas at Austin
dulgi7245@skku.edu, hwoo@skku.edu

## Abstract

Skill Incremental Learning (SIL) is the process by which an embodied agent expands and refines its skill set over time by leveraging experience gained through interaction with its environment or by the integration of additional data. SIL facilitates efficient acquisition of hierarchical policies grounded in reusable skills for downstream tasks. However, as the skill repertoire evolves, it can disrupt compatibility with existing skill-based policies, limiting their reusability and generalization. In this work, we propose SIL-C, a novel framework that ensures skill-policy compatibility, allowing improvements in incrementally learned skills to enhance the performance of downstream policies without requiring policy re-training or structural adaptation. SIL-C employs a bilateral lazy learning-based mapping technique to dynamically align the subtask space referenced by policies with the skill space decoded into agent behaviors. This enables each subtask, derived from the policy's decomposition of a complex task, to be executed by selecting an appropriate skill based on trajectory distribution similarity. We evaluate SIL-C across diverse SIL scenarios and demonstrate that it maintains compatibility between evolving skills and downstream policies while ensuring efficiency throughout the learning process. Source code: https://github.com/L2dulgi/SIL-C.

## 1 Introduction

Lifelong embodied agents must continuously integrate novel knowledge from unceasing streams of data into their evolving skill library while simultaneously leveraging previously acquired skills [1–4]. *Skill Incremental Learning* (SIL) supports this process by enabling agents to expand and refine their skill sets over time through continual interaction with the environment or integration of new data. Accordingly, SIL often aims to facilitate the development of hierarchical policies grounded in reusable skills for downstream tasks [5–7].

Recent research explores diverse formulations of SIL, such as modular skill composition [8–11], continual adaptation [12–14], and hierarchical policy learning [15, 16] to support scalable embodied agents. Yet, a critical challenge remains underexplored; as skills evolve over time, maintaining compatibility with downstream policies that depend on those skills becomes increasingly difficult [17–20]. Without proper alignment, updated skills may invalidate previously learned policies, limiting their reusability and generalization [21–24].

Figure 1 illustrates a policy-compatible SIL scenario involving two types of skill-policy compatibility: (i) *Forward Skill Compatibility (FwSC)*, which ensures that a newly added skill can be effectively utilized during the training of future downstream policies, and (ii) *Backward Skill Compatibility (BwSC)*, which ensures that existing downstream policies can continue to use newly added or updated skills without requiring re-training, and, when applicable, benefit from improved policy performance.

---

*Corresponding author: Honguk Woo (hwoo@skku.edu)

39th Conference on Neural Information Processing Systems (NeurIPS 2025).

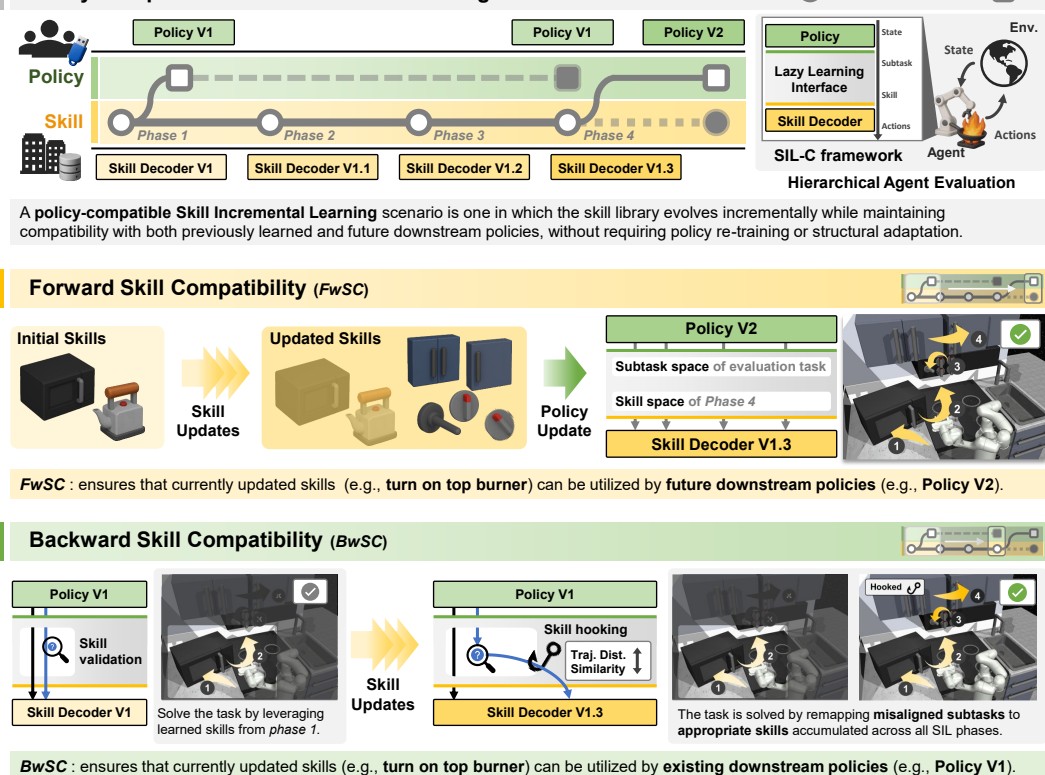

Figure 1: Overview of policy-compatible skill incremental learning and SIL-C framework

To address skill-policy compatibility challenges, we introduce SIL-C, a novel SIL framework with an interface layer, enabling hierarchical skill composition to seamlessly support downstream policy learning. For *FwSC*, inspired by append-only file systems, we design the framework to support non-destructive skill updates that mitigate forgetting and to introduce clear abstractions between task-specific policy learning and the task-agnostic evolving skill library, thereby enabling practical lifelong learning. For *BwSC*, we further design the interface within the framework to provide skill validation, which ensures that subtasks proposed by the high-level policy are contextually appropriate, and skill hooking, which intercepts misaligned subtasks and remaps them to the most suitable skill based on trajectory distribution similarity.

We implement both compatibility mechanisms through a lazy learning-based interface that maintains a consistent structure, enabling seamless integration with existing SIL approaches. To achieve *FwSC* and *BwSC* without requiring policy re-training or structural adaptation, we reformulate alignment between subtask space and skill space through trajectory distribution similarity as an instance-based classification problem with append-only prototype memories. By deferring decisions to inference time and offloading only the mapping function, this approach preserves existing training procedures while supporting the decoupled evolution of skills and policies.

With diverse SIL scenarios regarding compatibility, sample efficiency, and modularity, we evaluate SIL-C and other SIL baselines. SIL-C improves the normalized reward of downstream policies learned during the initial phase by an average of $18.6pp$ to $42.5pp$ across full scenarios, while the baselines only maintain or even degrade in performance. In particular, under the few-shot imitation setting in SIL, where each task is provided with only a single demonstration and 20% of its transitions, SIL-C achieves nearly double the overall performance of the baseline within the *BwSC* evaluation.

Our contributions are summarized as follows: (1) We propose SIL-C, a novel framework for SIL that maintains compatibility between evolving skills and downstream hierarchical policies. (2) We introduce a lazy learning-based mapping technique that aligns the subtask space from policies with the skill space based on trajectory distribution similarity. (3) We demonstrate the effectiveness of SIL-C across diverse SIL scenarios, showing that it maintains skill-policy compatibility while supporting efficient and scalable skill integration.

## 2    Related Work

**Skill incremental learning.** To achieve pre-trained behavior models for efficiently learning downstream tasks, prior work has explored unsupervised skill discovery [25, 26] and exploited task-agnostic data to build skill priors that can accelerate learning [27–30]. Contrastive objectives are also used to improve representation quality [31–34]. Several methods have been extended with new expert demonstrations to refine or expand priors for downstream adaptation [35–37]. Building on pre-trained models, recent work has investigated continual task adaptation in offline settings, aiming to handle streams of diverse tasks without full re-training [9, 38]. Beyond task-level adaptation, several approaches aim to incrementally acquire and organize skills at finer abstraction levels [39] by leveraging subgoal information [10, 11], often via hierarchical formulations [12, 13]. Yet, these SIL approaches typically assume synchronous updates between a skill library and downstream policies, limiting scalability when adding or removing skills.

**Continual updates for pre-trained models.** Expansion-based continual learning methods [40, 41] maintain task compatibility via explicit task identifiers, while recent adapter-based approaches [42–45] enable parameter-efficient updates. However, these typically rely on frozen backbones, limiting their ability to revise pre-trained knowledge [46]. Such issue arises in sequential decision-making [9, 11, 38], where pre-trained policies must adapt over time. Recently, a continual pre-training approach has been proposed to improve forward compatibility [47–49], but its integration into hierarchical settings remains underexplored. We explore this direction in the context of lifelong skill acquisition.

**Lazy learning for behavior learning.** Lazy learning techniques reduce the cost of online adaptation and mitigate forgetting by leveraging instance-based retrieval and local updates [50]. They have been applied in behavior learning through prototype memories and nearest-neighbor matching [51–53], and in robotics applications, particularly for adapting to novel tasks from limited demonstrations [54–59]. While effective for adaptation, these methods typically do not support long-term skill reuse or accumulation without additional training. We address this by extending lazy learning into the SIL setting, reformulating trajectory-similarity-based skill validation and hooking as an instance-based classification mechanism that enables skill reuse and composition without policy re-training.

## 3    Problem Formulation

We formulate each task $\tau$ as a Markov decision process (MDP) $\mathcal{M} = (\mathcal{S}, \mathcal{A}, \mathcal{P}, \mathcal{R}, \mu_0, \gamma)$, where $\mathcal{S}$ is the state space, $\mathcal{A}$ is the action space, $\mathcal{P}$ is the transition probability of the environment, $\mathcal{R}$ is the reward function, $\mu_0$ is the initial state distribution, and $\gamma$ is the discount factor. The objective is to learn a policy $\pi(a|s;\theta)$ that maximizes the expected return:

$$J(\theta) = \mathbb{E}_{\pi(a|s;\theta)} \left[ \sum_{t=0}^{\infty} \gamma^t \mathcal{R}(s_t, a_t) \right]. \tag{1}$$

To address long-horizon tasks, we adopt a hierarchical policy. A high-level policy $\pi_h(z_h|s; \theta_h)$ selects a subtask $z_h$, which is mapped to a skill $z_l = \psi(z_h)$ and executed by a low-level decoder $\pi_l(a|s, z_l; \theta_l)$. In our setup, the mapping $\psi$ may be the identity map or a more general transformation, with both $z_h$ and $z_l$ lying in a shared latent space $\mathcal{Z}$. The overall objective is denoted as $J(\theta_l, \theta_h)$ [60].

In *Skill Incremental Learning* (SIL), we observe a stream of datasets $\{\mathcal{D}_p\}_{p=1}^{P}$, where each $\mathcal{D}_p$ contains demonstrations for updating the skill decoder $\pi_l$ at phase $p$. The goal is to continually update the decoder with $\mathcal{D}_p$, while maintaining or improving performance across a set of evaluation tasks $\mathcal{T}_p$ presented at each phase. We assume access to task-level expert demonstrations $\mathcal{D}_\tau$ of $\tau$ for training the high-level policy $\pi_h$, enabling supervised learning and facilitating adaptation to evolving skill sets. The high-level policy for each task $\tau$ is updated independently of the phase and may be replaced or fixed. Our objective is to find the collections of high- and low-level optimal parameters$(\Theta_l^*, \Theta_h^*)$ over all phases, that maximize performance across all observed tasks:

$$(\Theta_l^*, \Theta_h^*) = \underset{\Theta_l, \Theta_h}{\arg\max} \, \mathbb{E}_{\leq p} \left[ \sum_{\tau \in \mathcal{T}_p} J_\tau(\theta_l^p, \theta_h^\tau) \right], \quad \Theta_l = \{\theta_l^p\}_{p=1}^{P}, \quad \Theta_h = \{\theta_h^\tau\}_{\tau \in \mathcal{T}}. \tag{2}$$

Here, $\theta_l^p$ and $\theta_h^\tau$ denote the parameters of the skill decoder $\pi_l^p$ at phase $p$ and the high-level policy $\pi_h^\tau$ for task $\tau$, respectively.

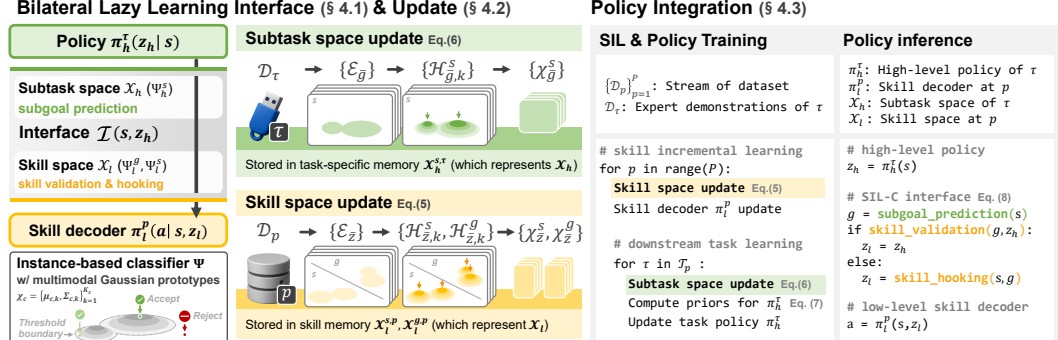

Figure 2: Overview of the SIL-C framework: components, updates, and integration

# 4 Our Framework

We introduce the SIL-C framework, a hierarchical architecture with a lazy learning interface designed to address skill-policy compatibility in SIL. It consists of a high-level policy $\pi_h^\tau$, a low-level skill decoder $\pi_l^p$, and their respective learning algorithms $A_h$ and $A_l$. The interface layer $\mathcal{I}$ connects the two levels by mapping each subtask $z_h$ to a skill $z_l$. Both subtask and skill are drawn from the shared index set $\bar{\mathcal{Z}}_p = \{1, \ldots, Z_p\}$, which expands over phases by appending new indices $\bar{\mathcal{Z}}_p = \{Z_{p-1}+1, \ldots, Z_p\}$. Given a state $s$, execution in SIL-C follows:

$$z_h \sim \pi_h^\tau(\cdot \mid s), \quad z_l = \mathcal{I}(s, z_h), \quad a \sim \pi_l^p(\cdot \mid s, z_l), \quad \text{where} \quad z_l, z_h \in \mathcal{Z}_p. \tag{3}$$

The core component of SIL-C is the bilateral lazy learning interface $\mathcal{I}$, which operates across two spaces: subtask space $\mathcal{X}_h$ and skill space $\mathcal{X}_l$. We implement this interface as a two-stage instance-based classifier that performs trajectory similarity matching between subtasks and skills. In our design, the subtask space $\mathcal{X}_h$ is represented by prototypes of subtask initial states used to predict the corresponding subgoal, while the skill space $\mathcal{X}_l$ is represented by prototypes that capture the distributions of initial states and subgoals for each skill. Together, these representations allow trajectory similarity to be compared across skills and subtasks. Building on this, interface operates in two stages: skill validation first checks whether the current subtask can achieve the task-specific subgoal predicted by $\mathcal{X}_h$ given the current state; if not, skill hooking remaps the subtask to the most feasible skill based on the current state. Figure 2 illustrates the components and their integration.

## 4.1 Bilateral Lazy Learning Interface

**Bilateral modules** ($\Psi_h^s, \Psi_l^g, \Psi_l^s$). To enable efficient trajectory distribution similarity matching under SIL, the interface $\mathcal{I}$ estimates similarity through bilateral lazy learning modules: the task-side module $\Psi_h^s$ operating on the subtask space $\mathcal{X}_h$ and the skill-side modules $\Psi_l^g, \Psi_l^s$ operating on the skill space $\mathcal{X}_l$. The two sides communicate via abstracted trajectory distributions represented by current state $s$ and subgoal $g$ pairs, where a subgoal represents a desired future state reachable from the current state $s$ within $m$ steps. To map subtask $z_h$ to an executable skill $z_l$, the task-side module $\Psi_h^s$ first predicts the target subgoal $g$ from $s$. Then, the skill-side module $\Psi_l^g$ assesses the direct executability of $z_h$ toward $g$, for skill validation. If validation fails, skill-side modules $\Psi_l^g, \Psi_l^s$ identify an appropriate skill via two-stage matching conditioned on $(s, g)$, for skill hooking.

**Instance-based classifier** ($\Psi$). To implement lazy learning in the interface, we adopt an instance-based classifier for all modules $\Psi$. Each label $c \in \mathcal{C}$, representing a concrete prediction target (e.g., a subgoal $g$ or a skill index $z$), is modeled by a multimodal Gaussian prototype $\chi_c = \{(\mu_{c,k}, \Sigma_{c,k})\}_{k=1}^{K_c}$, where $K_c$ denotes the number of Gaussian components for label $c$, and $\Sigma_{c,k}$ is diagonal for computational efficiency. For a query instance $x$, the module computes distances to all prototypes:

$$d_c(x) = \min_{k \le K_c} \sqrt{(x - \mu_{c,k})^\top \Sigma_{c,k}^{-1}(x - \mu_{c,k})}. \tag{4}$$

This Mahalanobis distance-based multimodal approach effectively captures complex distributions for both accurate prediction and reliable validation. Specifically, it enables two operations: (i) classification via $\Psi(x; \mathcal{C}) = \arg\min_{c \in \mathcal{C}} d_c(x)$ for nearest prototype selection, and (ii) validation via $\Psi(x, c) = \mathbb{I}[d_c(x) \le \delta_c]$ for out-of-distribution detection, where $\delta_c$ is the square root of 99% chi-square quantile [61].

## 4.2 Interface Update

We parameterize the interface modules with append-only prototype memories. On the task-side, classifier $\Psi_h^s$ relies on task-specific memory $\mathcal{X}_h^{s,\tau}$ to represent subtask space $\mathcal{X}_h$ for each task $\tau$. On the skill side, classifiers $\Psi_l^s$ and $\Psi_l^g$ use task-agnostic shared memories $\mathcal{X}_l^{s,p}$ and $\mathcal{X}_l^{g,p}$ to represent skill space $\mathcal{X}_l$ at phase $p$. The subtask space $\mathcal{X}_h$ is updated during downstream policy learning, while the skill space $\mathcal{X}_l$ is updated during SIL.

**Skill space update ($\mathcal{X}_l$).** At SIL phase $p$, SIL-C updates $\mathcal{X}_l$ by generating skill prototypes from the streamed dataset $\mathcal{D}_p$. An unsupervised skill clustering algorithm [62, 60] segments $\mathcal{D}_p$ into distinct skill groups $\{\mathcal{E}_{\bar{z}}\}_{\bar{z}\in\bar{\mathcal{Z}}_p}$, where $\bar{\mathcal{Z}}_p$ is the index set of newly discovered skills at phase $p$. The number of discovered skills $\bar{\mathcal{Z}}_p$ can either be fixed or automatically determined by the clustering algorithm. For each group $\mathcal{E}_{\bar{z}}$, we apply $K$-means separately to the state and subgoal spaces to capture the multimodal distribution within the group. The number of sub-clusters $K_{\bar{z}}$ is selected either manually or automatically via silhouette score [63]. This process yields sub-clusters $\{\mathcal{H}_{\bar{z},k}^s, \mathcal{H}_{\bar{z},k}^g\}_{k=1}^{K_{\bar{z}}}$ for each skill $\bar{z}$. Then, we compute $\mu_{\bar{z},k}$ and $\Sigma_{\bar{z},k}$ to create prototype for each skill, denoted as $\chi_{\bar{z}}^s$ and $\chi_{\bar{z}}^g$, respectively, corresponding to skill label $\bar{z}$. The overall process follows:

$$\underbrace{\mathcal{D}_p}_{\text{skill dataset}} \xrightarrow[\text{segment}]{\text{Skill Clustering}} \underbrace{\{\mathcal{E}_{\bar{z}}\}_{\bar{z}\in\bar{\mathcal{Z}}_p}}_{\text{skill groups}} \xrightarrow[s,g]{K\text{-means}} \underbrace{\{\mathcal{H}_{\bar{z},k}^s, \mathcal{H}_{\bar{z},k}^g\}_{\bar{z}\in\bar{\mathcal{Z}}_p}^{k\in[1:K_{\bar{z}}]}}_{\text{sub-clusters}} \xrightarrow[s,g]{\mu_{\bar{z},k},\Sigma_{\bar{z},k}} \underbrace{\{\chi_{\bar{z}}^s, \chi_{\bar{z}}^g\}_{\bar{z}\in\bar{\mathcal{Z}}_p}}_{\text{skill prototypes}}. \quad (5)$$

The resulting skill prototypes are stored in skill memory $\mathcal{X}_l^{s,p}$ and $\mathcal{X}_l^{g,p}$.

**Subtask space update ($\mathcal{X}_h$).** At downstream policy learning for task $\tau$, SIL-C constructs $\mathcal{X}_h$ by generating subtask prototypes from the most recent expert demonstrations $\mathcal{D}_\tau$, to the discretized subgoal groups $\mathcal{G}_\tau$. To efficiently represent demonstrations, we apply the same skill clustering algorithm [60] to derive $\mathcal{G}_\tau$, followed by the same sub-clustering procedure as in the skill space update:

$$\underbrace{\mathcal{D}_\tau}_{\text{expert dataset}} \xrightarrow[\text{segment}]{\text{Subtask Clustering}} \underbrace{\{\mathcal{E}_{\bar{g}}\}_{\bar{g}\in\mathcal{G}_\tau}}_{\text{subtask groups}} \xrightarrow[s]{K\text{-means}} \underbrace{\{\mathcal{H}_{\bar{g},k}^s\}_{\bar{g}\in\mathcal{G}_\tau}^{k\in[1:K_{\bar{g}}]}}_{\text{sub-clusters}} \xrightarrow[s]{\mu_{\bar{g},k},\Sigma_{\bar{g},k}} \underbrace{\{\chi_{\bar{g}}^s\}_{\bar{g}\in\mathcal{G}_\tau}}_{\text{subtask prototypes}}. \quad (6)$$

After prototype construction, subtask prototypes are stored in the task-specific memory $\mathcal{X}_h^{s,\tau}$.

## 4.3 Policy Integration

**Policy learning on the interface.** We utilize an energy-based prior [64] to guide the learning of the high-level policy $A_h$. This prior is induced by the skill decoder: given a state $s$, the decoder evaluates all candidate skill pairs $\{(s, z_l)\}_{z_l\in\mathcal{Z}_p}$ and assigns the subtask label whose decoded action best matches the expert action $a^*$. Specifically, we compute the subtask label as:

$$z_h = \underset{z_l\in\mathcal{Z}_p}{\arg\min}\|\hat{a} - a^*\|^2 \quad (7)$$

where $\hat{a} \sim \pi_l(\cdot \mid s, z_l)$ denotes the decoded action from the skill decoder using skill $z_l$. The decoder is continually updated by the skill incremental learning algorithm $A_l$.

**Policy inference via the interface.** The policy inference process follows the sequence defined in Eq. (3). Given a sampled subtask $z_h \sim \pi_h^\tau(\cdot \mid s)$ from the high-level policy, the interface first predicts the subgoal $g = \Psi_h^s(s; \mathcal{G}_\tau)$ using the task-side module, conditioned on the task-specific memory $\mathcal{X}_h^{s,\tau}$. Using this inferred subgoal $g$, the interface then performs skill validation by evaluating the distance of subgoal-subtask pair $(g, z_h)$ by skill-side module $\Psi_l^g(g, z_h)$. If the skill validation passes, the interface returns $z_l = z_h$. Otherwise, it initiates skill hooking, seeking an alternative skill better aligned with the inferred subgoal $g$. Formally, the decision rule for interface $\mathcal{I}$ follows:

$$\mathcal{I}(s, z_h) = \begin{cases} z_h, & \Psi_l^g(g, z_h) = 1 \\ \Psi_l^s(s; \mathcal{Z}'), & \Psi_l^g(g, z_h) = 0 \end{cases} \quad , \text{where} \quad g = \Psi_h^s(s; \mathcal{G}_\tau). \quad (8)$$

Here, candidate skill set $\mathcal{Z}' = \{z' \in \mathcal{Z}_p | \Psi_l^g(g, z') = 1\} \cup \{z_h\}$, where $z_h$, originally inferred subtask from the high-level policy, is included as a default fallback when no valid candidate skill exists.

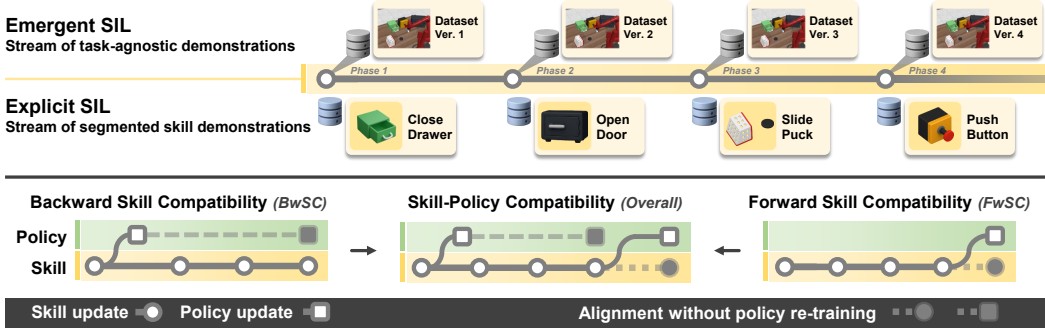

Figure 3: (*Top*) SIL scenario types and (*Bottom*) evaluation groups

# 5 Experiments

## 5.1 Experiment Settings

**Environments and SIL scenarios.** To evaluate skill-policy compatibility, we construct various SIL scenarios using two simulation environments: Franka Kitchen [65, 66] and Meta-World [67, 11]. In our experiments, we have four SIL phases where we add new skills by training the skill decoder, and train 24 downstream task policies using the updated skills for each phase.

As illustrated in the top of Figure 3, we define two scenario types to evaluate SIL under different levels of skill supervision. The *Emergent SIL* setting follows skill-based pretraining approaches [28, 27], introduces only task-agnostic demonstrations, reflecting an unsupervised SIL setup. In contrast, the *Explicit SIL* setting adopts the setup from prior SIL works [12, 68] and uses demonstrations that exhibit clearly chunked behaviors with predefined semantics.

As shown in the bottom of Figure 3, we categorize each scenario into three evaluation groups based on the combinations of skill decoders and policies used for task evaluation. The *BwSC group* pairs the skill decoder from each phase with the policy initially trained in the first phase. The *FwSC group* pairs the skill decoder from each phase with its synced policy trained using that decoder. The *Overall group* aggregates both *BwSC* and *FwSC* groups to measure comprehensive compatibility.

**Baselines.** We evaluate three baseline groups. **Type I** combines skill-based pre-training with continual learning strategies for SIL, adopting a simple skill appending scheme. We extend BUDS [62, 10] and PTGM [60], which discretize the skill space during pre-training. As continual learning methods, we apply fine-tuning (FT), experience replay (ER) [69], and adapter append (AA) [38], which trains an adapter after the initial phase and stores it for use in subsequent phases. **Type II** includes SIL approaches that use semantic goals as skill labels, relying on pre-defined supervision. They include prototype-based skill retrieval [11] and temporal replay with model expansion [12]. **Type III** corresponds to SIL-C, which uses the Type I configuration as its base architecture. We also report joint-training performance, obtained by training a single decoder on the union of all skill datasets.

All baselines adopt the same goal-conditioned skill decoder architecture [60], which conditions on subgoals hashed by skills, and use a diffusion-based model [70, 71] for decoding. For policy learning, they follow the same training objective, behavior cloning with a prior, as defined in Eq. (7).

**Metrics.** We measure skill-policy compatibility by adapting continual learning metrics from lifelong robot learning [4, 11, 38]. All task evaluation results are normalized to 100% based on the maximum reward defined in each environment. Forward Transfer (FWT) measures the performance of newly trained policies after skill updates, quantifying how updated skills facilitate learning new tasks. We report FWT at the *Initial* and *Final* phases of SIL. Backward Transfer (BWT) measures the change in performance from initial evaluation, assessing how skill updates affect previously evaluated tasks. Area Under the Curve (AUC) averages all phase performances within a scenario. All metrics are measured per task and reported as averages across tasks for each evaluation group in each scenario.

For comprehensive experiment settings including implementation details, hyperparameters, and additional configurations, please refer to the Appendix.

Table 1: Performance evaluation of skill-policy compatibility in two SIL scenarios of the Kitchen environment with four seeds. Each row corresponds to a baseline, categorized by the skill interface configuration, hierarchical agent structure, and the SIL algorithm used for skill decoder updates. The symbol $*$ denotes methods that require pre-defined semantic skill labels for both policy and decoder training. The left side reports *BwSC* results, the right side reports *FwSC* results, and the center shows the overall performance considering both. Best results are shown in **bold**. In this table, SIL-C uses the Type I (PTGM + AA) setting as its base architecture.

**Kitchen : *Emergent SIL***

| | Baselines | | *Initial* | *BwSC* (*Initial*) | | Overall performance | | *FwSC* (*Synced*) | | *Final* |
|---|---|---|---|---|---|---|---|---|---|---|
| Type | Skill Interface | SIL Algo. | FWT (%) | BWT (%) | AUC (%) | BWT (%) | AUC (%) | BWT (%) | AUC (%) | FWT (%) |
| I | Skill Segments [62, 10] (BUDS) | FT | $23.5_{\pm4.9}$ | $-16.4_{\pm5.8}$ | $11.2_{\pm1.2}$ | $-6.1_{\pm4.9}$ | $18.3_{\pm1.2}$ | $4.3_{\pm4.3}$ | $26.7_{\pm2.3}$ | $18.7_{\pm1.2}$ |
| | | ER | $23.5_{\pm3.8}$ | $-10.8_{\pm8.1}$ | $15.5_{\pm2.3}$ | $3.6_{\pm7.0}$ | $26.6_{\pm2.8}$ | $17.9_{\pm6.5}$ | $37.0_{\pm3.0}$ | $44.5_{\pm7.0}$ |
| | | AA | $21.9_{\pm4.1}$ | $1.3_{\pm4.4}$ | $22.9_{\pm4.0}$ | $15.7_{\pm3.5}$ | $35.3_{\pm2.9}$ | $30.1_{\pm3.9}$ | $44.4_{\pm2.6}$ | $57.1_{\pm7.5}$ |
| | Subgoal Bins [60] (PTGM) | FT | $45.0_{\pm2.0}$ | $-36.9_{\pm1.5}$ | $17.3_{\pm0.9}$ | $-24.3_{\pm1.6}$ | $24.1_{\pm1.0}$ | $-11.7_{\pm2.1}$ | $36.2_{\pm1.6}$ | $24.7_{\pm2.0}$ |
| | | ER | $46.8_{\pm2.5}$ | $-19.7_{\pm4.4}$ | $32.1_{\pm2.9}$ | $-5.0_{\pm2.8}$ | $42.5_{\pm1.4}$ | $9.7_{\pm3.3}$ | $54.0_{\pm1.8}$ | $57.7_{\pm3.6}$ |
| | | AA | $45.0_{\pm3.1}$ | $2.6_{\pm0.9}$ | $46.9_{\pm3.7}$ | $15.4_{\pm1.7}$ | $58.2_{\pm1.7}$ | $28.1_{\pm4.1}$ | $66.1_{\pm0.6}$ | $83.6_{\pm3.1}$ |
| II | Skill-prototypes [11]* Instructions [12]* | AA | $55.1_{\pm1.8}$ | $0.5_{\pm3.1}$ | $55.5_{\pm2.7}$ | $0.5_{\pm2.8}$ | $55.7_{\pm2.7}$ | $0.9_{\pm2.2}$ | $55.8_{\pm2.4}$ | $56.9_{\pm4.9}$ |
| | | ER | $54.0_{\pm2.4}$ | $18.5_{\pm4.0}$ | $\mathbf{67.8_{\pm1.1}}$ | $19.1_{\pm2.7}$ | $70.3_{\pm1.0}$ | $19.7_{\pm2.0}$ | $68.7_{\pm1.9}$ | $77.4_{\pm2.8}$ |
| III | SIL-C (w/ PTGM) | AA | $52.9_{\pm2.9}$ | $18.6_{\pm1.2}$ | $66.8_{\pm2.6}$ | $22.0_{\pm2.4}$ | $\mathbf{71.8_{\pm1.3}}$ | $25.4_{\pm4.0}$ | $\mathbf{71.9_{\pm0.9}}$ | $87.2_{\pm3.2}$ |
| - | Subgoal Bins [60] | Joint | - | - | - | - | - | - | - | $86.9_{\pm2.2}$ |

**Kitchen : *Explicit SIL***

| | Baselines | | *Initial* | *BwSC* (*Initial*) | | Overall performance | | *FwSC* (*Synced*) | | *Final* |
|---|---|---|---|---|---|---|---|---|---|---|
| Type | Skill Interface | SIL Algo. | FWT (%) | BWT (%) | AUC (%) | BWT (%) | AUC (%) | BWT (%) | AUC (%) | FWT (%) |
| I | Skill Segments [62, 10] (BUDS) | FT | $14.6_{\pm0.0}$ | $-13.6_{\pm1.0}$ | $4.4_{\pm0.8}$ | $-9.5_{\pm0.6}$ | $6.5_{\pm0.5}$ | $-5.4_{\pm0.6}$ | $10.5_{\pm0.4}$ | $3.0_{\pm1.9}$ |
| | | ER | $14.6_{\pm0.0}$ | $0.2_{\pm0.5}$ | $14.7_{\pm0.4}$ | $14.1_{\pm1.3}$ | $26.7_{\pm1.1}$ | $28.0_{\pm2.2}$ | $35.6_{\pm1.7}$ | $37.0_{\pm2.8}$ |
| | | AA | $14.6_{\pm0.0}$ | $0.0_{\pm0.0}$ | $14.6_{\pm0.0}$ | $20.9_{\pm1.1}$ | $32.5_{\pm1.0}$ | $41.9_{\pm2.3}$ | $46.0_{\pm1.7}$ | $62.8_{\pm3.0}$ |
| | Subgoal Bins [60] (PTGM) | FT | $14.6_{\pm0.0}$ | $-14.4_{\pm0.2}$ | $3.8_{\pm0.2}$ | $-8.9_{\pm1.0}$ | $7.0_{\pm0.9}$ | $-1.7_{\pm3.9}$ | $12.1_{\pm1.5}$ | $1.7_{\pm2.1}$ |
| | | ER | $14.6_{\pm0.3}$ | $-1.5_{\pm1.1}$ | $13.5_{\pm0.8}$ | $17.2_{\pm1.2}$ | $29.3_{\pm1.1}$ | $35.9_{\pm1.7}$ | $41.5_{\pm1.5}$ | $52.0_{\pm7.5}$ |
| | | AA | $14.5_{\pm0.2}$ | $0.0_{\pm0.1}$ | $14.6_{\pm0.4}$ | $25.8_{\pm1.0}$ | $36.7_{\pm0.8}$ | $51.6_{\pm1.9}$ | $\mathbf{53.3_{\pm1.4}}$ | $79.8_{\pm2.2}$ |
| II | Skill-prototypes [11]* Instructions [12]* | AA | $14.6_{\pm0.0}$ | $0.0_{\pm0.0}$ | $14.6_{\pm0.0}$ | $18.4_{\pm11.3}$ | $33.6_{\pm3.5}$ | $44.3_{\pm8.1}$ | $47.8_{\pm6.1}$ | $75.0_{\pm12.7}$ |
| | | ER | $14.6_{\pm0.0}$ | $-0.6_{\pm2.7}$ | $14.1_{\pm2.1}$ | $19.1_{\pm3.4}$ | $30.9_{\pm2.9}$ | $38.7_{\pm4.4}$ | $43.6_{\pm3.3}$ | $60.2_{\pm8.8}$ |
| III | SIL-C (w/ PTGM) | AA | $14.6_{\pm0.0}$ | $42.5_{\pm3.0}$ | $\mathbf{46.5_{\pm2.3}}$ | $46.2_{\pm2.5}$ | $\mathbf{54.1_{\pm2.1}}$ | $49.8_{\pm2.4}$ | $51.9_{\pm1.8}$ | $80.6_{\pm6.3}$ |
| - | Subgoal Bins [60] | Joint | - | - | - | - | - | - | - | $86.9_{\pm2.2}$ |

## 5.2 Skill-Policy Compatibility : Backward and Forward

Table 1 presents the skill-policy compatibility evaluation for the Franka Kitchen environment, covering both *Emergent* and *Explicit* skill incremental scenarios. It jointly reports *BwSC* and *FwSC*, and the overall performance shows that SIL-C achieves the highest AUC among all baselines.

**Backward Skill Compatibility (*BwSC*).** The left side of Table 1 reports the performance of the *Initial* phase policy after subsequent skill updates, evaluated across all phases. Compared to Type I baselines, SIL-C achieves a higher initial performance (*Initial* FWT) and closely matches the performance of methods with predefined skill labels. This suggests that SIL-C can robustly replace lower-confidence skills via skill validation and hooking. Furthermore, SIL-C matches the BWT and AUC scores of Type II methods. However, this is partly due to the nature of the *Emergent* scenario in Franka Kitchen, where future skills are already available and fully observable from the start, creating a near-oracle setting. In *Explicit* scenarios with incrementally revealed skill labels, SIL-C consistently improves performance, whereas most Type II methods show near-zero BWT, merely preserving initial performance without adaptation. Meanwhile, Type I baselines with limited replay suffer from skill forgetting, with only the *AA* variant maintaining backward compatibility but failing to improve it.

**Forward Skill Compatibility (*FwSC*).** The right side of Table 1 reports performance when policies are retrained at each phase to align with updated skills. SIL-C and PTGM with append-only skill learning achieve comparable final performance (*Final* FWT) to joint training, highlighting effective utilization of accumulated skills. It also achieves the highest AUC, indicating strong forward compatibility. Among Type I methods, ER and AA yield positive BWT, suggesting that newly acquired skills contribute to subsequent policy learning, although less effectively than SIL-C.

In Appendix C, we provide additional overviews and extend the experiments with diverse configurations of SIL algorithms to further motivate our design choices, and present experiments with varying SIL phase orderings, showing that SIL-C consistently maintains skill-policy compatibility.

Table 2: Few-shot imitation results on Kitchen *Emergent* SIL. **Shots** denotes expert demonstrations per task for high-level policy training. **Ratio** indicates the proportion of transitions uniformly sampled from each demonstration. All baselines use SIL algorithm AA for skill decoder. For SIL-C at Ratio = 50% and 20%, each sampled transition is treated as an individual subtask prototype.

| Kitchen : *Emergent SIL* | | | | | | | | | | | |
|---|---|---|---|---|---|---|---|---|---|---|---|
| **Baselines** | | | | *Initial* | *BwSC (Initial)* | | Overall performance | | *FwSC (Synced)* | | *Final* |
| Type | Skill Interface | Shots | Ratio | FWT (%) | BWT (%) | AUC (%) | BWT (%) | AUC (%) | BWT (%) | AUC (%) | FWT (%) |
| I | Subgoal Bins [60] (PTGM) | 5 | 100% | 40.5±0.9 | 0.4±2.4 | 40.8±2.1 | 10.0±1.4 | 49.1±2.0 | 19.6±2.0 | 55.3±2.4 | 67.6±6.0 |
| | | 3 | 100% | 34.7±2.1 | 0.7±4.9 | 35.1±2.5 | 5.9±4.8 | 38.6±2.0 | 11.0±5.6 | 40.2±1.0 | 45.7±3.6 |
| | | 1 | 100% | 26.5±1.9 | 1.7±2.5 | 27.7±2.2 | 6.1±2.4 | 31.7±0.9 | 10.5±3.4 | 34.3±0.9 | 37.8±1.9 |
| | | 1 | 50% | 28.6±5.7 | -0.8±4.4 | 27.9±3.2 | 1.0±5.1 | 29.5±1.8 | 2.9±6.0 | 30.8±1.2 | 36.5±4.0 |
| | | 1 | 20% | 25.7±3.5 | -2.9±2.0 | 23.5±2.2 | 1.5±4.8 | 27.0±1.2 | 5.9±7.6 | 30.1±2.5 | 30.5±4.4 |
| III | SIL-C (w/ PTGM) | 5 | 100% | 47.4±3.2 | 11.3±2.4 | 55.9±4.6 | 17.5±1.2 | 62.4±2.7 | 23.6±4.2 | 65.1±2.3 | 75.8±4.4 |
| | | 3 | 100% | 44.4±1.8 | 11.2±2.5 | 52.8±2.9 | 14.9±2.2 | 57.2±1.8 | 18.6±3.3 | 58.4±1.2 | 68.5±4.0 |
| | | 1 | 100% | 43.1±5.4 | 11.9±2.8 | 52.0±5.3 | 15.7±3.1 | 56.5±3.7 | 19.4±4.8 | 57.6±2.9 | 65.7±3.6 |
| | | 1 | 50% | 41.6±4.8 | 11.7±2.3 | 50.4±4.7 | 14.9±2.6 | 54.3±2.7 | 18.0±4.7 | 55.1±1.6 | 67.2±3.5 |
| | | 1 | 20% | 37.3±6.4 | 12.0±3.3 | 46.4±5.7 | 14.4±5.6 | 49.7±2.6 | 16.8±9.9 | 49.9±3.0 | 58.5±8.1 |

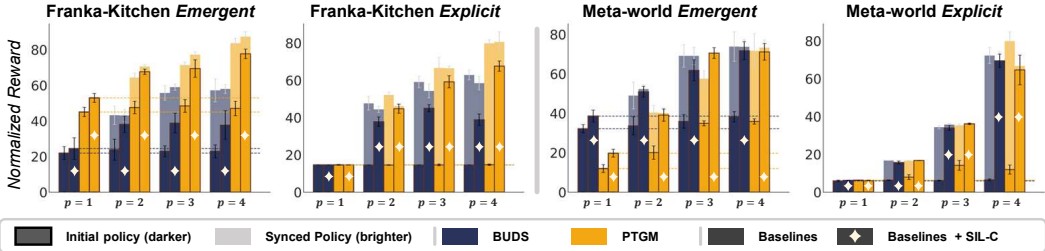

Figure 4: Results on Kitchen and Meta-World tasks under *Emergent* and *Explicit* skill incremental scenarios. Each group of bars represents a skill update phase ($x$-axis), and the $y$-axis shows the scaled reward. Darker bars represent evaluation with the initial policies. Lighter bars show performance after re-training with updated skills. Diamonds ($\diamondsuit$) indicate the application of SIL-C to each baseline. The skill decoder uses the AA strategy for Kitchen and ER for Meta-World.

## 5.3 Sample Efficiency: Downstream Few-shot Imitation Learning

**Effect of number of demonstrations.** Table 2 shows that across all demonstration budgets, SIL-C outperforms the baseline, highlighting the benefit of skill validation and hooking via lazy learning under limited supervision. With 5 demonstrations per task (5-shots), SIL-C improves overall performance AUC from 49.1% to 62.4% and *Final* FWT from 67.6% to 75.8%, indicating utility even with ample supervision. When using only 1 demonstration per task (1-shot), the gains become more significant, with AUC increasing from 31.7% to 56.5% and *Final* FWT from 37.8% to 65.7%.

**Effect of transition sampling ratio.** Reducing the number of transitions extracted from demonstrations (from 100% to 20%) has only a moderate effect on SIL-C, which still maintains a positive BWT and achieves an AUC above 46% on *BwSC*. In contrast, the baseline degrades substantially, with AUC dropping below 24%. This indicates that the skill-conditioned interface provides robustness to sparse supervision by substituting missing transitions with reusable behaviors. These results suggest that the interface enables strong generalization by leveraging prior skills in low-data regimes.

## 5.4 Modularity: Under Varying Design Choices for Hierarchical Architecture

Figure 4 shows that across all scenarios, SIL-C consistently improves *Initial Policy* performance by enhancing *BwSC*, regardless of environment, skill clustering strategy, or SIL algorithm for low-level decoder. In the Franka Kitchen environment, as shown in Table 1, SIL-C enables both BUDS and PTGM with AA to make more effective use of their accumulated skills, leading to more efficient reuse. This effect is clearly reflected in the visualized performance. In the Meta-World environment, unlike in Kitchen, both BUDS and PTGM use ER to train a shared skill decoder. While this promotes generalization and enables reuse of new skills without policy re-training, the actual gains remain limited. When SIL-C is applied, we observe a clear improvement in evaluation with the initial policy, suggesting that skill validation and hooking are effective in filtering unreliable skills. Although SIL-C relies less on broadly generalized skills from ER, it tends to reuse verified skills more precisely, which may contribute to safer and more stable policy execution.

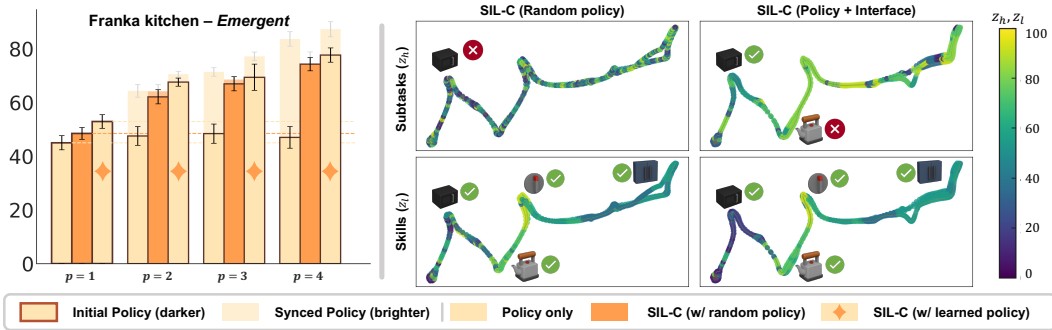

Figure 5: Ablation of the SIL-C lazy learning interface in the Kitchen *Emergent SIL* scenario using PTGM with AA configuration. (*Left*) Per-phase performance across three settings. (*Right*) Evaluation trajectories example illustrating selected subtasks and corresponding skills used to solve the task: `open microwave` → `move kettle` → `turn on top burner` → `open hinge cabinet`.

Table 3: Performance under varying input noise levels in Kitchen *Emergent SIL* scenario. All baselines use the same SIL algorithm AA for the decoder.

**Kitchen : *Emergent SIL***

| | | | *Initial* | *BwSC* (*Initial*) | | Overall performance | | *FwSC* (*Synced*) | | *Final* |
|---|---|---|---|---|---|---|---|---|---|---|
| Type | Skill Interface | Noise | FWT (%) | BWT (%) | AUC (%) | BWT (%) | AUC (%) | BWT (%) | AUC (%) | FWT (%) |
| I | Subgoal Bins [60] (PTGM) | ×1 | $45.0_{\pm3.1}$ | $2.6_{\pm0.9}$ | $46.9_{\pm3.7}$ | $15.4_{\pm1.7}$ | $58.2_{\pm1.7}$ | $28.1_{\pm4.1}$ | $66.1_{\pm0.6}$ | $83.6_{\pm3.1}$ |
| | | ×2 | $44.4_{\pm4.5}$ | $0.8_{\pm2.0}$ | $45.0_{\pm3.4}$ | $10.7_{\pm2.7}$ | $53.5_{\pm2.2}$ | $20.5_{\pm3.9}$ | $59.7_{\pm1.9}$ | $68.6_{\pm3.5}$ |
| | | ×3 | $35.8_{\pm5.3}$ | $0.5_{\pm3.7}$ | $36.1_{\pm2.9}$ | $3.2_{\pm5.6}$ | $38.5_{\pm1.5}$ | $5.9_{\pm7.7}$ | $40.2_{\pm1.7}$ | $40.5_{\pm4.6}$ |
| | | ×5 | $19.7_{\pm1.7}$ | $-1.2_{\pm3.4}$ | $18.8_{\pm1.7}$ | $-1.8_{\pm2.4}$ | $18.2_{\pm0.9}$ | $-2.4_{\pm2.5}$ | $17.9_{\pm1.1}$ | $15.9_{\pm1.6}$ |
| III | SIL-C (w/ PTGM) | ×1 | $52.9_{\pm2.9}$ | $18.6_{\pm1.2}$ | $66.8_{\pm2.6}$ | $22.0_{\pm2.4}$ | $71.8_{\pm1.3}$ | $25.4_{\pm4.0}$ | $71.9_{\pm0.9}$ | $87.2_{\pm3.2}$ |
| | | ×2 | $49.7_{\pm0.6}$ | $11.3_{\pm1.8}$ | $58.2_{\pm1.5}$ | $16.7_{\pm1.1}$ | $64.0_{\pm1.3}$ | $22.1_{\pm1.9}$ | $66.3_{\pm1.8}$ | $76.4_{\pm2.6}$ |
| | | ×3 | $35.6_{\pm2.2}$ | $13.6_{\pm4.2}$ | $45.8_{\pm3.1}$ | $15.4_{\pm2.9}$ | $48.8_{\pm1.5}$ | $17.3_{\pm3.4}$ | $48.5_{\pm1.0}$ | $52.9_{\pm3.9}$ |
| | | ×5 | $19.6_{\pm2.3}$ | $4.3_{\pm1.3}$ | $22.8_{\pm2.4}$ | $3.8_{\pm2.4}$ | $22.9_{\pm1.4}$ | $3.3_{\pm4.1}$ | $22.1_{\pm1.4}$ | $23.1_{\pm4.1}$ |

## 6 Ablation and Analysis

### 6.1 Lazy Learning Interface Ablation

Figure 5 ablates lazy learning interface and policy integration contributions across three settings: policy-only, SIL-C with a random policy, and SIL-C with a learned policy. While the policy-only configuration cannot leverage newly added skills without retraining. In contrast, the SIL-C w/ random policy shows that skill validation and hooking mechanisms alone offer minimal decision-making capability via trajectory distribution similarity matching, but still underperform due to limited generalization ability compared to using a learned high-level policy. The SIL-C w/ learned policy further improves performance by using skill validation with out-of-distribution detection to correct subtask-skill mismatches, which is crucial for error-sensitive long-horizon tasks. Evaluation trajectories show that while the random policy fails immediately from inconsistent subtask selection and the policy-only configuration struggles with the not-yet-acquired skill `move kettle`, SIL-C completes tasks through the interface remapping subtasks to utilize newly added skills.

### 6.2 Robustness

Table 3 evaluates robustness in the Kitchen environment by scaling input noise during evaluation, where the noise parameter [65] is increased from ×1 to ×5. SIL-C consistently outperforms PTGM+AA across all noise levels. Notably, under ×5 noise, SIL-C maintains a positive BWT of 4.3%. The performance gap further increases with larger skill library sizes. For example, under ×3 noise, the FWT difference grows from a marginal $-0.2\%$ *Initial* FWT to $12.4\%$ *Final* FWT at convergence with 80 skills, where SIL-C achieves $52.9\%$ compared to $40.5\%$ for PTGM+AA.

This robustness stems from architectural differences in handling noisy observations. When noise corrupts high-level policy outputs, the bilateral interface in SIL-C performs skill validation to detect out-of-distribution subtasks and skill hooking to remap them based on trajectory similarity. In contrast, the static skill mapping in PTGM+AA cannot adapt to distributional shifts, leading to error accumulation in long-horizon tasks. The widening performance gap suggests that trajectory-based matching becomes increasingly effective as the skill repertoire expands.

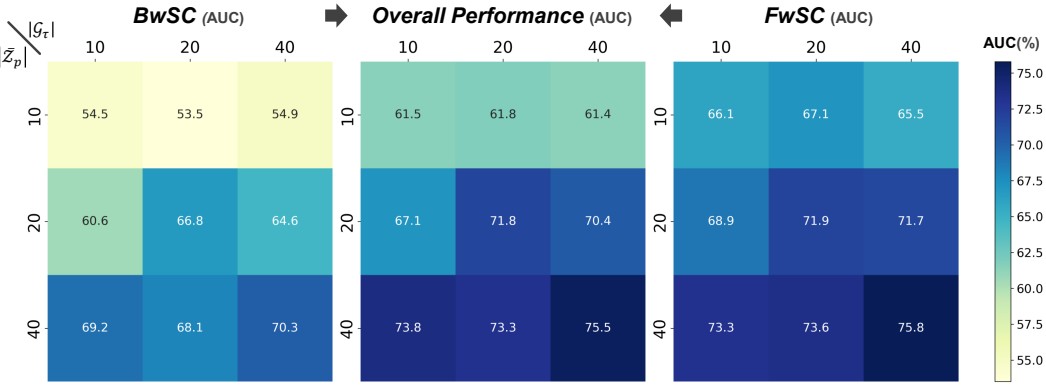

Figure 6: Analysis with varying $|\bar{\mathcal{Z}}_p|$ and $|\mathcal{G}_\tau|$ of SIL-C (w/ PTGM, *AA*) in Kitchen *Emergent SIL* scenario. Rows represent different values of $|\bar{\mathcal{Z}}_p|$ while columns represent different values of $|\mathcal{G}_\tau|$. AUC results shown for (*Left*) *BwSC*, (*Center*) Overall performance, and (*Right*) *FwSC*.

## 6.3 Skill and Subtask Space Resolution

Figure 6 examines how the resolution of skill and subtask spaces influences *FwSC* and *BwSC*. We vary the number of skill groups per phase ($|\bar{\mathcal{Z}}_p|$) and subtask groups per task ($|\mathcal{G}_\tau|$) in SIL-C (with PTGM and AA). Starting from the default setting in Table 1 ($|\bar{\mathcal{Z}}_p| = 20$, $|\mathcal{G}_\tau| = 20$), we test doubled and halved values to assess sensitivity to space resolution, which is determined by the granularity and quality of unsupervised clustering. Full results are reported in Appendix D.1.

**Skill space resolution** ($|\bar{\mathcal{Z}}_p|$). Increasing $|\bar{\mathcal{Z}}_p|$ from 10 to 40 raises overall AUC from 61.5% to 75.5% when $|\mathcal{G}_\tau| = 40$. This improvement is consistent across all subtask settings, with $|\bar{\mathcal{Z}}_p| = 40$ yielding the best performance in both *BwSC* (70.3%) and *FwSC* (75.8%). Higher skill space resolution also reduces dependence on subtask space resolution. With $|\bar{\mathcal{Z}}_p| = 40$, *BwSC* remains stable across subtask configurations (69.2%–70.3%), indicating that sufficient skill diversity compensates for coarse subtask space.

**Subtask space resolution** ($|\mathcal{G}_\tau|$). The effect of subtask space resolution depends on skill space resolution. With coarse skill space ($|\bar{\mathcal{Z}}_p| = 10$), increasing $|\mathcal{G}_\tau|$ from 10 to 40 changes performance only marginally (61.5% to 61.4% in *Overall Performance*). In contrast, with finer skill space ($|\bar{\mathcal{Z}}_p| = 40$), the same increase yields gains (73.8% to 75.5%). This shows that sufficient skill diversity is necessary for benefiting from finer subtask space, as coarse skill space limits the effect of additional subtask prototypes.

## 7 Conclusion

We present SIL-C, a framework for policy-compatible skill incremental learning (SIL) that introduces a bilateral lazy learning interface to preserve *FwSC* and *BwSC* without requiring full re-training or structural adaptation of downstream policies. Empirical results demonstrate that, by ensuring skill-policy compatibility, SIL-C consistently improves performance, efficiency, and modularity.

**Future Directions.** While our approach performs well in the simulation-based environments with well-defined state representations, future work may explore settings with more diverse or noisy skill distributions, where unsupervised clustering alone becomes less effective. Enhancing robustness in such conditions and leveraging minimal goal information for exploration can improve sample efficiency. Additionally, monitoring skill reliability and distribution shifts during deployment may help enforce expected behaviors and detect anomalies, contributing to safer execution.

**Impact Statement.** Our framework aims to support large-scale deployment in practical settings where new skills must be integrated without disrupting existing behaviors. By maintaining policy compatibility and enabling decentralized skill sharing, SIL-C provides a foundation for scalable robotic systems that can evolve over time. This line of research has potential to assist the development of adaptive, modular agents in applied domains.

# Acknowledgement

This work was supported by Institute of Information & communications Technology Planning & Evaluation (IITP) grant funded by the Korea government (MSIT), (RS-2022-00143911, AI Excellence Global Innovative Leader Education Program, RS-2022-II220043 (2022-0-00043), Adaptive Personality for Intelligent Agents, RS-2022-II221045 (2022-0-01045), Self-directed multi-modal Intelligence for solving unknown, open domain problems, RS-2025-02218768, Accelerated Insight Reasoning via Continual Learning, and RS-2025-25442569, AI Star Fellowship Support Program(Sungkyunkwan Univ.) RS-2019-II190421, Artificial Intelligence Graduate School Program (Sungkyunkwan University)), the National Research Foundation of Korea (NRF) grant funded by the Korea government (MSIT) (No. RS-2023-00213118), IITP-ITRC (Information Technology Research Center) grant funded by the Korea government (MIST) (IITP-2025-RS-2024-00437633, 10%), IITP-ICT Creative Consiliece Program grant funded by the Korea government (MSIT) (IITP-2025-RS-2020-II201821, 10%), and by Samsung Electronics.

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

# Appendix

## Contents

# A SIL-C Implementation

This section details key notations in A.1 and operational steps in A.2 and throughout the method. We aim to facilitate understanding of concepts described in the main text and supplementary explanations provided in this appendix.

## A.1 Notations

This subsection summarizes all notations used throughout the paper.

Table 4: Notation used in the paper

| Symbol | Meaning | Symbol | Meaning |
|---|---|---|---|
| **MDP fundamentals** | | | |
| $\mathcal{M}$ | $(\mathcal{S},\mathcal{A},\mathcal{P},\mathcal{R},\mu_0,\gamma)$ | $\mathcal{S}$ | State space |
| $\mathcal{A}$ | Action space | $\mathcal{P}$ | Transition probability |
| $\mathcal{R}$ | Reward function | $\mu_0$ | Initial-state distribution |
| $\gamma$ | Discount factor $(0 < \gamma < 1)$ | $s$ | State |
| $a$ | Action | $r,\ R$ | Step reward; episode return |
| $\pi$ | (Stochastic) policy | $\tau$ | Task identifier |
| **Hierarchical policy** | | | |
| $\pi_h$ | High-level policy | $\pi_l$ | Low-level skill decoder |
| $z_h$ | High-level subtask index | $z_l$ | Low-level skill index |
| $\theta_h$ | Parameters of $\pi_h$ | $\theta_l$ | Parameters of $\pi_l$ |
| $\mathcal{Z}$ | Discrete subtask/skill index space | | |
| **Skill Incremental Learning (SIL)** | | | |
| $p$ | SIL phase index | $P$ | Total # of phases |
| $\{\mathcal{D}_p\}_{p=1}^P$ | Datastream (task-agnostic trajectories) | $\mathcal{T}_p$ | Evaluation task set at phase $p$ |
| $\mathcal{Z}_p$ | Subtask/skill index set at phase $p$ | $\mathcal{D}_\tau$ | Expert demonstrations for task $\tau$ |
| $\pi_h^\tau$ | High-level policy for task $\tau$ | $\pi_l^p$ | Skill decoder at phase $p$ |
| $\theta_h^\tau$ | Params of $\pi_h$ for task $\tau$ | $\theta_l^p$ | Params of $\pi_l$ at phase $p$ |
| **Lazy-Learning Interface (bilateral modules & instance based classifier)** | | | |
| $\mathcal{X}_h$ | Subtask space | $\mathcal{X}_l$ | skill space |
| $g$ | Subgoal state | $m$ | Steps ahead to define $g$ |
| $\Psi_h^s(s;\mathcal{G}_\tau)$ | *Task-side* subgoal predictor: returns $g$ | $\mathcal{X}_h^{s,\tau}$ | Task memory of $\tau$ (subtask prototypes) |
| $\Psi_l^g(g,z)$ | *Skill-side* validator: 1 iff $d_z(g) \le \delta_z$ | $\mathcal{X}_l^{g,p}$ | Skill memory at phase $p$ (subgoal-based prototypes) |
| $\Psi_l^s(s;\mathcal{Z}')$ | *Skill-side* classifier (restricted to $\mathcal{Z}'$): returns $z_l$ | $\mathcal{X}_l^{s,p}$ | Skill memory at phase $p$ (state-based prototypes) |
| $\mathcal{I}(s,z_h)$ | Interface (Eq. (8)) | $\mathcal{Z}'$ | Candidate skills s.t. $\Psi_l^g(g,z')=1$ |
| $\delta_c$ | Distance threshold for class $c$ | $d_c(x)$ | Mahalanobis distance to class-$c$ prototype |
| $\chi_c$ | Prototype for class $c$ | $\mu_{c,k},\Sigma_{c,k}$ | Mean and (diagonal) covariance of $k$-th Gaussian |
| $K_c$ | # of sub-clusters (modes) for $\chi_c$ | $x$ | Query instance (e.g., $s$ or $g$) |
| **Prototype construction (skill & subtask spaces)** | | | |
| $\mathcal{E}_{\bar{z}}$ | Clustered skill group | $\bar{\mathcal{Z}}_p$ | Index set of skills discovered at phase $p$ |
| $\mathcal{H}_{\bar{z},k}^s,\ \mathcal{H}_{\bar{z},k}^g$ | Sub-clusters for skill $\bar{z}$ (state/subgoal) | $K_{\bar{z}}$ | # of sub-clusters for skill $\bar{z}$ |
| $\chi_{\bar{z}}^s,\ \chi_{\bar{z}}^g$ | Prototypes for skill $\bar{z}$ (state/subgoal) | $\mu_{\bar{z},k},\Sigma_{\bar{z},k}$ | Mean, covariance of skill-$\bar{z}$ sub-cluster |
| $\mathcal{E}_{\bar{g}}$ | Clustered subtask group | $\mathcal{G}_\tau$ | Subtask label set (from demos) |
| $\mathcal{H}_{\bar{g},k}^s$ | Sub-clusters for subtask of subgoal $\bar{g}$ (state) | $K_{\bar{g}}$ | # of sub-clusters for subtask of subgoal $\bar{g}$ |
| $\chi_{\bar{g}}^s$ | Subtask prototype (state) | $\mu_{\bar{g},k},\Sigma_{\bar{g},k}$ | Mean, covariance of subtask sub-cluster of subgoal $\bar{g}$ |

## A.2 Algorithms

We provide phase-level training and run-time inference procedures that instantiate the SIL-C framework using the append-only prototype memories and the bilateral lazy-learning interface described in Sec. 4.

---

**Algorithm 1** SIL-C on SIL & Policy Learning with Interface at phase $p$

---

**Require:** Phase index $p$, streamed dataset $\mathcal{D}_p$, set of evaluation tasks $\mathcal{T}_p$, demonstrations $\{\mathcal{D}_\tau\}_{\tau \in \mathcal{T}_p}$,
    previous decoder $\pi_l^{(p-1)}$, previous memories $\mathcal{X}_l^{s,(p-1)}, \mathcal{X}_l^{g,(p-1)}$, previous index set $\mathcal{Z}_{p-1}$
**Ensure:** Updated decoder $\pi_l^p$, updated memories $\mathcal{X}_l^{s,p}, \mathcal{X}_l^{g,p}$, updated index set $\mathcal{Z}_p$, trained high-
    level policies $\{\pi_h^\tau\}_{\tau \in \mathcal{T}_p}$, task memories $\{\mathcal{X}_h^{s,\tau}\}_{\tau \in \mathcal{T}_p}$

 # SIL AT PHASE-$p$
 # SKILL SPACE UPDATE (EQ. (5))                  ▷
 **Skill clustering**: segment $\mathcal{D}_p$ into $\{\mathcal{E}_{\bar{z}}\}_{\bar{z} \in \bar{\mathcal{Z}}_p}$
 Initialize $\bar{\mathcal{X}}_l^{s,p} \leftarrow \emptyset$, $\bar{\mathcal{X}}_l^{g,p} \leftarrow \emptyset$
 **for all** $\bar{z} \in \bar{\mathcal{Z}}_p$ **do**
  **Sub-clustering**: apply K-means to $\mathcal{E}_{\bar{z}}$ to obtain $\{\mathcal{H}_{\bar{z},k}^s, \mathcal{H}_{\bar{z},k}^g\}_{k=1}^{K_{\bar{z}}}$
  **Prototype creation**: compute $\{\mu_{\bar{z},k}, \Sigma_{\bar{z},k}\}_{k=1}^{K_{\bar{z}}}$ and form $\chi_{\bar{z}}^s, \chi_{\bar{z}}^g$
  **Add prototypes**: $\bar{\mathcal{X}}_l^{s,p} \leftarrow \bar{\mathcal{X}}_l^{s,p} \cup \{\chi_{\bar{z}}^s\}$,   $\bar{\mathcal{X}}_l^{g,p} \leftarrow \bar{\mathcal{X}}_l^{g,p} \cup \{\chi_{\bar{z}}^g\}$
 **Memory append**: $\mathcal{X}_l^{s,p} \leftarrow \mathcal{X}_l^{s,(p-1)} \cup \bar{\mathcal{X}}_l^{s,p}$,   $\mathcal{X}_l^{g,p} \leftarrow \mathcal{X}_l^{g,(p-1)} \cup \bar{\mathcal{X}}_l^{g,p}$

 # LOW-LEVEL SKILL DECODER UPDATE              ▷
 Train $\pi_l$ with $A_l$ on $\mathcal{D}_p$ to obtain $\pi_l^p$
 **Index growth**: $\mathcal{Z}_p \leftarrow \mathcal{Z}_{p-1} \cup \bar{\mathcal{Z}}_p$

 # DOWNSTREAM POLICY TRAINING AT PHASE-$p$
 **for all** $\tau \in \mathcal{T}_p$ **do**
  # SUBTASK SPACE UPDATE (EQ. (6))              ▷
  Initialize $\mathcal{X}_h^{s,\tau} \leftarrow \emptyset$
  **Subtask clustering**: segment $\mathcal{D}_\tau$ into $\{\mathcal{E}_{\bar{g}}\}_{\bar{g} \in \mathcal{G}_\tau}$
  **for all** $\bar{g} \in \mathcal{G}_\tau$ **do**
   **Sub-clustering**: apply K-means to $\mathcal{E}_{\bar{g}}$ to obtain $\{\mathcal{H}_{\bar{g},k}^s\}_{k=1}^{K_{\bar{g}}}$
   **Prototype creation**: compute $\{\mu_{\bar{g},k}, \Sigma_{\bar{g},k}\}_{k=1}^{K_{\bar{g}}}$ and form $\chi_{\bar{g}}^s$
   **Add prototypes**: $\mathcal{X}_h^{s,\tau} \leftarrow \mathcal{X}_h^{s,\tau} \cup \{\chi_{\bar{g}}^s\}$

  # LABEL ASSIGNMENT WITH UPDATED DECODER (EQ. (7))     ▷
  **for all** transition $(s, a^*) \in \mathcal{D}_\tau$ **do**
   $z_h^* \leftarrow \arg\min_{z_l \in \mathcal{Z}_p} \|\hat{a} - a^*\|_2^2$,    $\hat{a} \sim \pi_l^p(\cdot \mid s, z_l)$

  # HIGH-LEVEL POLICY OPTIMIZATION             ▷
  Update $\pi_h^\tau$ by minimizing $\mathbb{E}_{(s,z_h^*)}[-\log \pi_h^\tau(z_h^* \mid s)]$
 **return** $\left( \pi_l^p, \mathcal{X}_l^{s,p}, \mathcal{X}_l^{g,p}, \mathcal{Z}_p, \{\pi_h^\tau\}_{\tau \in \mathcal{T}_p}, \{\mathcal{X}_h^{s,\tau}\}_{\tau \in \mathcal{T}_p} \right)$

---

**Phase training (Alg. 1).** *(A) Skill space update.* At the beginning of phase $p$, the streamed dataset $\mathcal{D}_p$ is segmented into skill groups $\{\mathcal{E}_{\bar{z}}\}_{\bar{z} \in \bar{\mathcal{Z}}_p}$ ((5)). For each group, we perform K-means in the state and subgoal spaces to obtain sub-clusters $\{\mathcal{H}_{\bar{z},k}^s, \mathcal{H}_{\bar{z},k}^g\}$ and form Gaussian prototypes $\chi_{\bar{z}}^s, \chi_{\bar{z}}^g$ with diagonal covariance. Newly created prototypes are *appended* to the skill memories $\mathcal{X}_l^{s,p}, \mathcal{X}_l^{g,p}$ without altering past entries, and the skill decoder is updated to $\pi_l^p$; the index set grows append-only as $\mathcal{Z}_p \leftarrow \mathcal{Z}_{p-1} \cup \bar{\mathcal{Z}}_p$. *(B) Policy learning on the interface.* For each task $\tau$, we build the subtask memory by clustering $\mathcal{D}_\tau$ into $\{\mathcal{E}_{\bar{g}}\}_{\bar{g} \in \mathcal{G}_\tau}$ and constructing state-side prototypes $\{\chi_{\bar{g}}^s\}$ ((6)). Labels for behavior cloning are assigned by the energy-based prior that selects $z_h^*$ minimizing the decoder mismatch ((7)), and $\pi_h^\tau$ is trained with cross-entropy over $(s, z_h^*)$.

**Algorithm 2** SIL-C Evaluation with Interface

---

Decoder $\pi_l^p$, policy $\pi_h^\tau$, interface modules $\Psi_l^{s,p}$, $\Psi_l^{g,p}$, $\Psi_h^{s,\tau}$,
task memory $\mathcal{G}_\tau$, skill index set $\mathcal{Z}_p$, initial state $s_0$
Initialize $R \leftarrow 0$, $s \leftarrow s_0$
**while** episode not terminated **do**

> \# HIGH-LEVEL POLICY
> $z_h \sim \pi_h^\tau(z_h \mid s)$                                   ▷ sample subtask
>
> \# INTERFACE INFERENCE (SUBTASK SPACE)
> $g \leftarrow \Psi_h^{s,\tau}(s; \mathcal{G}_\tau)$                                 ▷ predict subgoal
>
> \# INTERFACE INFERENCE (SKILL SPACE) (EQ. (8))
> **if** $\Psi_l^{g,p}(g, z_h) = 1$ **then**                          ▷ skill validation
> > $z_l \leftarrow z_h$
>
> **else**                                             ▷ skill hooking
> > $\mathcal{Z}' \leftarrow \{ z' \in \mathcal{Z}_p \mid \Psi_l^{g,p}(g, z') = 1 \} \cup \{z_h\}$
> > $z_l \leftarrow \Psi_l^{s,p}(s; \mathcal{Z}')$
>
> \# LOW-LEVEL SKILL DECODER
> $a \sim \pi_l^p(a \mid s, z_l)$                                   ▷ sample action
> Execute $a$; observe reward $r$ and next state $s'$; $R \leftarrow R + r$; $s \leftarrow s'$

**return** $R$

---

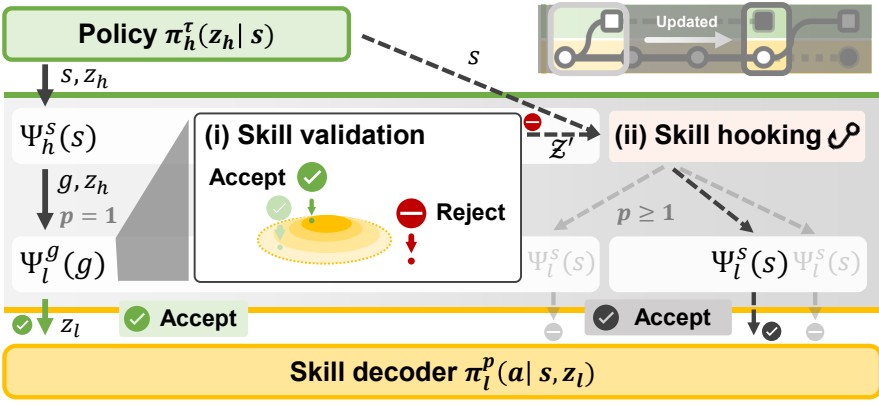

Figure 7: Visualization of agent inference procedure with SIL-C

**Agent Inference (Algoritm 2, Figure 7).** Given the current state $s$, the high-level policy samples a subtask $z_h \sim \pi_h^\tau(z \mid s)$, and the task-side module predicts a subgoal $g = \Psi_h^s(s; \mathcal{G}_\tau)$. The interface first performs *skill validation* with the skill-side validator $\Psi_l^g$: if $\Psi_l^g(g, z_h) = 1$, the subtask is accepted ($z_l = z_h$); otherwise, *skill hooking* restricts the candidate set to $\mathcal{Z}' = \{ z' \in \mathcal{Z}_p \mid \Psi_l^g(g, z') = 1 \} \cup \{z_h\}$ and chooses $z_l = \Psi_l^s(s; \mathcal{Z}')$ (*restricted classification*). Equivalently, validation can be written as $d_z(g) \leq \delta_z$ using the Mahalanobis distance in Eq. (4). This two-stage instance-based procedure operationalizes Eq. (8) and realizes compatibility at inference without policy re-training.

## A.3 Pseudocode

```python
## Initialization
from collections import defaultdict
from scipy.stats import chi2

Interface = {
    "skills":   defaultdict(lambda: {"subgoal": [], "state": []}),
    "subtasks": defaultdict(lambda: defaultdict(list)),
} # Gaussian component = (mean, precision)
pi_l, pi_h = None, {}

def label_dist(x, comps): # Eq.(4)
    def maha(x, comp):
        mu, Sigma_inv = comp
        d = x - mu
        return d.T @ Sigma_inv @ d
    return min(maha(x, c) for c in comps)

def chi2_thr(comps, conf=0.99):
    dim = comps[0][0].shape[0]
    return chi2.ppf(conf, dim)

## Policy compatible skill incremental learning
for p in range(1, P + 1):
    # Skill space update Eq.(5)
    chi_gz, chi_sz = create_skill_prototypes(cluster_skills(D_p))
    for z in chi_gz:
        Interface["skills"][z]["subgoal"].extend(chi_gz[z])
        Interface["skills"][z]["state"].extend(chi_sz[z])
    # Skill incremental learning
    pi_l = train_low_level_decoder(D_p, Interface["skills"], pi_l)

    # Downstream policy learning
    for tau in T_p:
        # Subtask space update Eq.(6)
        chi_g = create_subtask_prototypes(cluster_subtasks(D_tau))
        for g in chi_g:
            Interface["subtasks"][tau][g].extend(chi_g[g])

        # Policy learning on the interface Eq.(7)
        labels = assign_subtask_labels(D_tau, Interface["skills"], pi_l)
        pi_h[tau] = train_high_level_policy(D_tau, labels)

## Policy inference via the interface
def infer_action(s, tau, conf=0.99):
    # High-level policy: subtask proposal
    z_h = pi_h[tau](s)
    g   = predict_subgoal(s, Interface["subtasks"][tau])
    delta = chi2_thr(Interface["skills"][z_h]["subgoal"], conf)

    # Interface: skill validation and skill hooking Eq.(8)
    if label_dist(g, Interface["skills"][z_h]["subgoal"]) <= delta:
        z_l = z_h
    else:
        cand = [z for z in Interface["skills"]
            if label_dist(g, Interface["skills"][z]["subgoal"]) <= delta] + [z_h]
        z_l  = min(cand, key=lambda z:
            label_dist(s, Interface["skills"][z]["state"]))

    # Low-level decoder: action execution
    return pi_l(s, z_l)
```

Figure 8: Python style pseudo code of SIL-C

We provide Python-style pseudo code in Figure 8 to illustrate the SIL-C implementation. The code shows how the append-only prototype memories are constructed and updated during each phase, and how the bilateral interface performs skill validation and hooking at inference time using Mahalanobis distance matching.

## A.4 Hyperparameters

We apply a consistent set of hyperparameters to update each space in the SIL-C interface for both the Kitchen and Meta-World environments. The configuration, used in Table 1, is summarized in Table 5.

Table 5: Default hyperparameter configuration for SIL-C

| Skill-Side (per phase) | | Task-Side (per task) | |
|---|---|---|---|
| Sub-clusters per skill ($K_{\bar{z}}$) | 4 | Sub-clusters per sub-task ($K_{\bar{g}}$) | 4 |
| Goal offset ($m$) | 20 | Goal offset ($m$) | 20 |

# B Experimental Settings

## B.1 Environments

**Franka Kitchen** [65] is a long-horizon manipulation environment where a robot must complete multi-stage tasks by interacting with various kitchen objects such as a `microwave`, `kettle`, `burners`, `light switches`, and `cabinet doors`. We use the `kitchen-mixed-v0` dataset for experiment. Each demonstration consists solely of state-action transitions, without any reward signals or task labels. The state space is 60-dimensional, encompassing both the robot arm state and the states of the manipulated objects. The action space is 9-dimensional, corresponding to the 9 degrees of freedom (DoF) of the Franka robotic arm. Each task requires the robot to complete a sequence of 4 subtasks(stage) selected from a fixed set of 7 predefined subtasks: `open microwave`, `move kettle`, `turn on top burner`, `turn on bottom burner`, `light switch`, `open hinge cabinet`, `open slide cabinet`. A reward of 1.00 is given for each subtask successfully completed in the correct order, yielding a maximum of 4.00 when all subtasks are completed. For evaluation, we normalize the reward such that the maximum score is 100.

**Multi-stage Meta-World** extends the original Meta-World benchmark [67], which includes 50 diverse single-step robot tasks, by composing multiple tasks into sequential stages within a single episode. Each multi-stage task requires an agent to complete 4 subtasks in a fixed order, mimicking realistic long-horizon objectives `puck`, `drawer`, `button`, `door`, `box`, `handle`, `lever`, `stick` [77]. We use the `Easy` dataset introduced by [11], which consists of composed tasks involving 4 objects selected from a subset of: `puck`, `drawer`, `button`, `door`, with varied subtask sequences. Each demonstration consists solely of state-action transitions, without any reward signals or task labels. The state space is 140-dimensional, capturing both the robot arm state and the states of the relevant objects. The action space is 4-dimensional, corresponding to the 4 degrees of freedom of the robotic arm. Each task consists of 4 sequential subtasks selected from the predefined set: `slide puck`, `close drawer`, `push button`, `open door`, `close box`, `press handle`, `pull lever`, and `insert stick into red box`. As in the Franka Kitchen environment, a reward of 1.00 is given for each subtask completed in the correct order, with a maximum cumulative reward of 4.00 per task. For evaluation, we normalize the reward so that the maximum possible score is 100.

## B.2 SIL Scenario

In each phase $p$, a new skill dataset $\mathcal{D}_p$ is provided to train the skill decoder. Subsequently, based on the trained low-level skill decoder, 24 task-specific high-level policies are individually trained.

Table 6: Kitchen: *Emergent SIL*

| Phase | Skill Dataset | | | |
|---|---|---|---|---|
| 1 | microwave | bottom burner | light switch | slide cabinet |
| | bottom burner | top burner | light switch | slide cabinet |
| | microwave | bottom burner | top burner | light switch |
| | microwave | bottom burner | slide cabinet | hinge cabinet |
| | microwave | bottom burner | top burner | slide cabinet |
| | bottom burner | top burner | slide cabinet | hinge cabinet |
| | kettle | bottom burner | top burner | slide cabinet |
| | microwave | bottom burner | top burner | hinge cabinet |
| 2 | microwave | kettle | bottom burner | hinge cabinet |
| | microwave | kettle | top burner | hinge cabinet |
| | kettle | bottom burner | top burner | hinge cabinet |
| | microwave | kettle | top burner | light switch |
| | microwave | top burner | light switch | hinge cabinet |
| | microwave | kettle | light switch | hinge cabinet |
| 3 | microwave | light switch | slide cabinet | hinge cabinet |
| | microwave | kettle | slide cabinet | hinge cabinet |
| | microwave | kettle | bottom burner | slide cabinet |
| | microwave | kettle | light switch | slide cabinet |
| 4 | kettle | top burner | light switch | slide cabinet |
| | kettle | bottom burner | slide cabinet | hinge cabinet |
| | kettle | bottom burner | light switch | hinge cabinet |
| | kettle | bottom burner | top burner | light switch |
| | kettle | light switch | slide cabinet | hinge cabinet |
| | kettle | bottom burner | light switch | slide cabinet |

Table 7: Meta-World: *Emergent SIL*

| Phase | Skill Datasets | | | |
|---|---|---|---|---|
| 1 | puck | drawer | button | door |
| | puck | drawer | door | button |
| | puck | button | drawer | door |
| | puck | button | door | drawer |
| | puck | door | drawer | button |
| | puck | door | button | drawer |
| 2 | drawer | puck | button | door |
| | drawer | puck | door | button |
| | drawer | button | puck | door |
| | drawer | button | door | puck |
| | drawer | door | puck | button |
| | drawer | door | button | puck |
| 3 | button | puck | drawer | door |
| | button | puck | door | drawer |
| | button | drawer | puck | door |
| | button | drawer | door | puck |
| | button | door | puck | drawer |
| | button | door | drawer | puck |
| 4 | door | puck | drawer | button |
| | door | puck | button | drawer |
| | door | drawer | puck | button |
| | door | drawer | button | puck |
| | door | button | puck | drawer |
| | door | button | drawer | puck |

Table 8: Kitchen: *Explicit SIL*

| Phase | Skill Datasets |
|---|---|
| 1 | microwave |
| 2 | kettle / bottom burner |
| 3 | top burner / light switch |
| 4 | slide cabinet / hinge cabinet |

Table 9: Meta-World: *Explicit SIL*

| Phase | Skill Datasets |
|---|---|
| 1 | puck |
| 2 | drawer |
| 3 | button |
| 4 | door |

### B.2.1 Datastream Types : Emergent and Explicit SIL

***Emergent SIL.*** In the *Emergent SIL* scenario, we divide each dataset into four datastreams by grouping full task-agnostic demonstrations, resulting in datastreams that contain a mixture of tasks without explicit task identifiers. To facilitate scenario construction and explanation, we partition the dataset using task information. To maintain the task-agnostic nature of skill learning, this information is not exposed during skill-incremental learning, and the learning process remains fully task-agnostic. In the case of **Franka Kitchen**, we collect task demonstrations based on the predefined subtask goal information provided by the environment. Failed or truncated trajectories are discarded, and we identify 24 distinct tasks. The resulting datastream composition is summarized in Table 6. For **Meta-World**, we follow a similar approach, segmenting the dataset into 24 tasks, corresponding to all $4!$ possible subtask sequence permutations, by verifying subtask goal completion using the environment's built-in success conditions. The datastream configuration is shown in Table 7.

***Explicit SIL.*** In the *Explicit SIL* scenario, we segment each dataset into shorter demonstrations based on predefined skill boundaries specified by the environment. These skill segments are then grouped into four datastreams. To facilitate scenario construction and explanation, we use skill labels provided by the environment to organize the data. However, this information is not exposed to the agent during training, and the learning process remains fully skill-agnostic. For **Franka Kitchen**, we segment trajectories by identifying transitions that correspond to each predefined subtask. These are clustered

by subtask type to form skill-specific datastreams, as shown in Table 8. For **Meta-World**, we apply a similar segmentation strategy, isolating the trajectory from the beginning of each subtask to the point where the agent returns to a neutral position. These segments are then clustered according to subtask identity, resulting in the skill incremental setup summarized in Table 9.

### B.2.2 Evaluation Groups for SIL Scenarios

At each phase $p$, a new skill dataset $D_p$ is used to train the skill decoder. Using this updated decoder, we then train 24 task-specific high-level policies, one for each evaluation task. Across both environments, we evaluate all 24 tasks listed in Table 6 and Table 7 by training a dedicated policy for each task. This setup enables us to evaluate bidirectional compatibility. Specifically, we measure *Backward Skill Compatibility* (BwSC) by checking whether policies trained in earlier phases remain functional with updated skills. In parallel, we assess *Forward Skill Compatibility* (FwSC) by determining whether new policies can effectively leverage skills learned in previous phases.

Formally, the evaluation groups for $\text{BwSC}_\tau$ and $\text{FwSC}_\tau$ of task $\tau$ are defined as:

$$\text{BwSC}_\tau = [(\pi_h^{\tau,(1)}, \pi_l^{(1)}), (\pi_h^{\tau,(1)}, \pi_l^{(2)}), \dots, (\pi_h^{\tau,(1)}, \pi_l^{(p)})] \tag{9}$$

$$\text{FwSC}_\tau = [(\pi_h^{\tau,(1)}, \pi_l^{(1)}), (\pi_h^{\tau,(2)}, \pi_l^{(2)}), \dots, (\pi_h^{\tau,(p)}, \pi_l^{(p)})] \tag{10}$$

Here, $\pi_h^{\tau,(1)}$ denotes the initial high-level policy trained with the skill decoder from the first phase $p = 1$. For any phase $p \geq 1$, $\pi_h^{\tau,(p)}$ refers to the updated high-level policy synchronized with the corresponding skill decoder $\pi_l^{(p)}$ at phase $p$.

### B.3 Baseline Details

We categorize our baselines into four distinct types based on their methodology and compatibility with the Skill Incremental Learning (SIL) scenario.

**Type I: Skill-based approaches with continual learning.** Type I combines skill-based pre-training with continual learning strategies using a simple skill-space appending scheme. We adopt two hierarchical skill pre-training methods:

- **BUDS** [62, 10]: BUDS [62] segments trajectories into fixed-length intervals and merges them bottom-up based on trajectory similarity, and was later adopted as the foundation in LOTUS [10] for continual imitation learning. It discovers skills by clustering these segments. In our implementation, we base our hyperparameter choices on those reported in BUDS. To support goal-conditioned skill decoder learning, we annotate each trajectory with subgoal states prior to skill discovery. For each discovered skill, we select its goal representation as the subgoal state closest to the average of the subgoals observed across its transitions. Each phase can generate up to 10 skills, but the actual number is determined automatically using the silhouette score. In the Kitchen environment, this resulted in an average of approximately 8 skills per phase.
- **PTGM** [60]: PTGM leverages subgoal state information during skill decoding by discretizing the subgoal state space and treating each subgoal bin as a distinct skill. This approach supports pre-training in open domains and demonstrates strong performance on downstream tasks, particularly in complex environments, often outperforming skill pre-training methods based on continuous skill representations. In our experiments, we adopt the hyperparameters from the PTGM paper for the Kitchen domain. Each phase produces a total of 20 skill clusters.

In both methods, we define the subgoal state for each transition as the state reached after $m = 20$ steps. To support the SIL scenario, we expand the skill set at each phase by appending newly discovered skills to those from previous phases. This expansion is strictly append-only and does not modify previously learned skills.

These pre-trained skills are integrated with continual learning or adaptation strategies as follows:

- **Fine-Tuning (FT)**: The simplest approach. The skill decoder is incrementally trained on each new skill without access to prior data, no memory or replay is used.

- **Experience Replay (ER)** [69]: In our implementation, a replay buffer stores 10% of the data from each previous phase. During training in the next phase, these stored samples are interleaved with new data at a 1:1 sampling ratio to mitigate forgetting.
- **Adapter Append (AA)** [38]: In the first phase, we pre-train the entire model. Starting from the second phase, we freeze the pre-trained model and train a separate LoRA adapter [78] for each new phase, using rank 4 for the *Emergent* scenario and rank 16 for the *Explicit* scenario. During evaluation, we use the adapter corresponding to the phase in which the skill was introduced. This strategy, introduced in TAIL [38], preserves skill-policy compatibility by preventing forgetting in both the base model and its adapted parameters. However, it limits the ability to transfer or incorporate newly acquired skills across different phases.

**Type II: Semantic representation-based skill incremental learning approaches.** These methods rely on predefined semantic subgoals as skill labels. They incorporate prototype-based skill retrieval and model expansion with temporal replay.

- **Skill-prototypes + (AA)** [11]: Performs skill retrieval using a prototype memory, where each prototype is linked with adapter parameters and indexed by persistent semantic skill labels. We set the number of skill prototype bases to 50. Following the original setup, tasks are learned using semantic subgoal labels provided in a fixed sequence.
- **Instructions + (ER)** [12]: Defines and learns skills incrementally from semantic representations. We adopt its trajectory-based temporal replay to support skill incremental learning.

To ensure a fair comparison, these methods are restricted to using only the semantically labeled skills available for policy training at the corresponding SIL phase $p$. Trajectories without semantic skill labels are merged with the skill from the previous transition.

**Type III: SIL-C.** Our method builds on the Type I structure and its configuration. In our SIL setup, we assign four sub-clusters to each skill prototype. For subtask prototype construction, we follow the PTGM algorithm, setting 20 prototypes per task and assigning four sub-clusters to each prototype as well.

**High-level policy and skill decoder implementation.** For all baselines, we implement the high-level policy as a four-layer MLP that acts as a classifier for sub-task prediction. To enable behavior cloning in SIL-C, we generate supervision signals using Eq. (7) and train the classifier with cross-entropy loss.

The skill decoder, used across all baselines, is a goal-conditioned policy consisting of two components. The first maps each skill to a hashed embedding. Following the skill decoder architecture proposed in [60], we compute each skill embedding as the subgoal state closest to the centroid of the subgoal states comprising that skill. We then feed this subgoal state, along with the current state, into a conditional denoising diffusion model with four conditioned denoising blocks. The model denoises from Gaussian noise to reconstruct the corresponding action.

For Type II methods, where skills are defined semantically (e.g., using natural language descriptions), we embed the descriptions using the `text-embedding-3-large` model from OpenAI and use these vectors directly as skill representations.

Table 10: Default policy configuration

| Hyperparameter | Value |
|---|---|
| Model | MLP |
| Hidden size | 512 |
| Hidden layers | 4 |
| Dropout | 0.1 |
| State input dim | 60 |
| Optimizer | Adam |
| Learning rate | $1 \times 10^{-4}$ |
| $\beta_1$ | 0.9 |

Table 11: Default skill decoder configuration

| Hyperparameter | Value |
|---|---|
| Diffusion Model | DDPM [70] |
| Denoising step | 64 |
| Sampler(inference) | Deterministic (no noise added) |
| Schedule | Linear |
| Linear start | 1e-4 |
| Linear end | 2e-2 |
| Block | MLP |
| Hidden dimension | 512 |
| Layers | 4 |
| Dropout | 0.0 |
| Clip denoised | True |
| Optimizer | Adam |
| Learning rate | $2 \times 10^{-5}$ |
| Learning rate (LoRA) | $1 \times 10^{-4}$ |
| $\beta_1$ | 0.9 |

## B.4 Metrics

For each scenario, we evaluate the *FwSC* and *BwSC* groups using three metrics: Forward Transfer (FWT), Backward Transfer (BWT), and Area Under the Curve (AUC). *Overall performance* is reported by computing each metric over the union of both groups. Let $r^{\tau,p}$ denote the normalized reward for task $\tau$ at phase $p$, evaluated using the corresponding high-level policy and skill decoder. For example, $r^{\tau,p} = (\pi_h^{\tau,(1)}, \pi_l^p)$ for the *BwSC* group, and $r^{\tau,p} = (\pi_h^{\tau,p}, \pi_l^p)$ for the *FwSC* group. This score reflects task performance under the given high-level and low-level policy configuration. We assume a fixed evaluation task set $\mathcal{T}_1$, which is used across all phases, i.e., $\mathcal{T}_p = \mathcal{T}_1$ for all $p \in \{1, \ldots, P\}$.

**Forward Transfer (FWT)** measures the performance of newly trained policies after skill updates, quantifying how updated skills facilitate learning new tasks.

$$\text{FWT}^{\tau,p} = r^{\tau,p} \tag{11}$$

We report both the *Initial FWT*, computed at $p = 1$, and the *Final FWT*, computed at the final phase $p = P$.

**Backward Transfer (BWT)** quantifies the influence of skill incremental learning on previously acquired task performance. It is defined as the average change in performance across phases, relative to phase 1:

$$\text{BWT}^{\tau} = \frac{1}{P-1} \sum_{p=2}^{P} \left( r^{\tau,p} - r^{\tau,(1)} \right) \tag{12}$$

**Area Under the Curve (AUC)** captures the overall performance trend across all skill learning phases, computed as the mean normalized reward:

$$\text{AUC}^{\tau} = \frac{1}{P} \sum_{p=1}^{P} r^{\tau,p} \tag{13}$$

We report the final values for each metric by averaging over all evaluated tasks $\tau \in \mathcal{T}$.

## B.5 Training Details

**Compute Resources** We conducted our experiments in the following computing environments:

- AMD Ryzen 9 7950X3D 16-Core Processor with a single RTX 4090 GPU. OS: Ubuntu 22.04, CUDA Version: 12.4, Driver Version: 550.144.03
- AMD Ryzen Threadripper PRO 5975WX 32-Core CPU with 2× RTX 4090 GPUs. OS: Ubuntu 22.04, CUDA Version: 12.4, Driver Version: 550.144.03

- AMD Ryzen Threadripper PRO 5975WX 32-Core CPU with 2× RTX 4090 GPUs. OS: Ubuntu 22.04, CUDA Version: 12.2, Driver Version: 535.230.02

**Training Framework Details** We implemented our framework using JAX [79] to efficiently accelerate training and handle the continual phase's alternating training and evaluation process. All software dependencies are included with the released code for full reproducibility. The versions of core libraries are:

- `jax`: 0.4.34
- `jaxlib`: 0.4.34
- `flax`: 0.10.2
- `optax`: 0.1.9

**Experiment Compute Estimates** For experiments on Kitchen and Meta-World, we ran 4 seeds in parallel (4 processes on a single GPU). Each run took approximately 4 hours and 30 minutes, broken down as follows:

- **Training**: 1 hour for training 4 skill update phases, covering a total of 96 policy training runs (24 tasks per phase).
- **Evaluation**: 3 hours and 30 minutes for 168 task evaluations (24 tasks × [3 *BwSC* + 4 *FwSC*]).

**Total Compute Usage**

- **Main experiments only:** Approximately 225 GPU hours (RTX 4090), including core experiments and additional SIL scenario evaluations.
- **Experiments including Appendix:** Approximately 500 GPU hours (RTX 4090), covering extended experiments and additional SIL evaluations.
- **All experiments, including preliminary and discarded runs:** An estimated 600 GPU hours (RTX 4090), accounting for exploratory and failed runs not included in the final results.

### B.6 Evaluation Details

To report performance metrics, we used *four* random seeds and report the mean and standard deviation across them. For each scenario, the normalized reward of a given task was computed by running the evaluation three times and averaging the results. The averaged value was recorded as the reward for that task.

Table 12: Performance evaluation of skill-policy compatibility in two SIL scenarios of the Franka-Kitchen environment with four seeds. Each row corresponds to a baseline, categorized by the skill interface configuration, hierarchical agent structure, and the SIL algorithm used for skill decoder updates. The symbol $*$ denotes methods that require pre-defined semantic skill labels for both policy and decoder training. The left side reports *BwSC* results, the right side reports *FwSC* results, and the center shows the overall performance considering both.

**Kitchen : *Emergent SIL***

| Baselines | | | *Initial* | *BwSC* (*Initial*) | | Overall performance | | *FwSC* (*Synced*) | | *Final* |
|---|---|---|---|---|---|---|---|---|---|---|
| Type | Skill Interface | SIL Algo. | FWT (%) | BWT (%) | AUC (%) | BWT (%) | AUC (%) | BWT (%) | AUC (%) | FWT (%) |
| I | Skill Segments [62, 10] (BUDS) | FT | 23.5±4.9 | -16.4±5.8 | 11.2±1.2 | -6.1±4.9 | 18.3±1.2 | 4.3±4.3 | 26.7±2.3 | 18.7±1.2 |
| | | ER | 23.5±3.8 | -10.8±8.1 | 15.5±2.3 | 3.6±7.0 | 26.6±2.8 | 17.9±6.5 | 37.0±3.0 | 44.5±7.0 |
| | | AA | 21.9±4.1 | 1.3±4.4 | 22.9±4.0 | 15.7±3.5 | 35.3±2.9 | 30.1±3.9 | 44.4±2.6 | 57.1±7.5 |
| | Subgoal Bins [60] (PTGM) | FT | 45.0±2.0 | -36.9±1.5 | 17.3±0.9 | -24.3±1.6 | 24.1±1.0 | -11.7±2.1 | 36.2±1.6 | 24.7±2.0 |
| | | ER (10%) | 46.8±2.5 | -19.7±4.4 | 32.1±2.9 | -5.0±2.8 | 42.5±1.4 | 9.7±3.3 | 54.0±1.8 | 57.7±3.6 |
| | | ER (50%) | 48.2±1.9 | -9.2±2.2 | 41.3±2.0 | 5.8±1.8 | 53.1±1.8 | 20.8±1.9 | 63.8±1.7 | 74.7±2.2 |
| | | MT (100%) | 48.4±1.4 | -3.1±1.8 | 46.1±0.9 | 11.6±0.9 | 58.4±1.2 | 26.2±0.9 | 68.1±1.7 | 81.3±4.5 |
| | | AA | 45.0±3.1 | 2.6±0.9 | 46.9±3.7 | 15.4±1.7 | 58.2±1.7 | 28.1±4.1 | 66.1±0.6 | 83.6±3.1 |
| II | Skill-prototypes [11]* Instructions [12]* | AA | 55.1±1.8 | 0.5±3.1 | 55.5±2.7 | 0.5±2.8 | 55.7±2.7 | 0.9±2.2 | 55.8±2.4 | 56.9±4.9 |
| | | ER | 54.0±2.4 | 18.5±4.0 | 67.8±1.1 | 19.1±2.7 | 70.3±1.0 | 19.7±2.0 | 68.7±1.9 | 77.4±2.8 |
| III | SIL-C | MT (100%) | 52.7±2.8 | 15.2±2.8 | 64.1±2.6 | 19.7±2.1 | 69.6±2.1 | 24.3±1.7 | 70.9±1.7 | 82.7±0.7 |
| | | AA | 52.9±2.9 | 18.6±1.2 | 66.8±2.6 | 22.0±2.4 | 71.8±1.3 | 25.4±4.0 | 71.9±0.9 | 87.2±3.2 |
| - | Sub-goal Bins [60] | Joint | - | - | - | - | - | - | - | 86.9±2.2 |

# C  Additional Experiments

## C.1  Motivating Examples

Table 12 extends our main results with additional baselines to demonstrate a critical challenge in skill incremental learning: even multi-task learning, often considered an oracle baseline in continual learning, fails to achieve backward skill compatibility (*BwSC*).

**Multi-task learning fails to enable compositional skill reuse.** We evaluate PTGM with varying replay ratios (10%, 50%, 100%) to understand the limits of experience replay in SIL. Even with 100% replay, where the model has access to all previous data, PTGM achieves only -3.1% BWT in the BwSC setting. This negative transfer indicates that despite having full access to historical data, the method cannot leverage newly acquired skills to improve existing policies without complete retraining. The pattern persists across all replay configurations: ER (10%) shows -19.7% BWT, ER (50%) shows -9.2% BWT, and even multi-task learning (MT 100%) exhibits negative transfer. These results reveal a fundamental architectural limitation: naive combinations of continual learning approaches with skill-based pre-training fail to support dynamic skill composition.

**SIL-C enables true compositional learning without retraining.** In contrast, SIL-C achieves substantial positive backward transfer across both continual learning strategies: 15.2% BWT with MT (100%) and 18.6% BWT with AA. This improvement is not merely incremental but represents a qualitative difference in capability. While baseline methods require complete policy retraining to utilize new skills (as evidenced by their improved *FwSC* but poor *BwSC* performance), SIL-C's lazy learning interface enables existing policies to dynamically compose newly acquired skills at inference time. This compositional property is particularly evident in the overall performance metrics, where SIL-C maintains competitive or superior AUC scores on *BwSC* (64.1% with MT, 66.8% with AA) while simultaneously achieving the highest final performance (82.7% and 87.2% FWT respectively).

The significance of these results extends beyond numerical improvements: they demonstrate that SIL-C fundamentally changes how skills and policies interact in hierarchical reinforcement learning, enabling true lifelong learning where each new skill enhances the agent's entire behavioral repertoire without the computational burden of retraining.

## C.2  Skill-Policy Compatibility : Overview and Ordering Effects

Figure 9 presents phase-wise normalized rewards across four SIL datastreams, each defined by a different permutation of skill datasets from Table 1 (**Seq. 1**). The corresponding evaluation metrics for each stream are summarized in Table 13. These experiments assess both backward skill compatibility

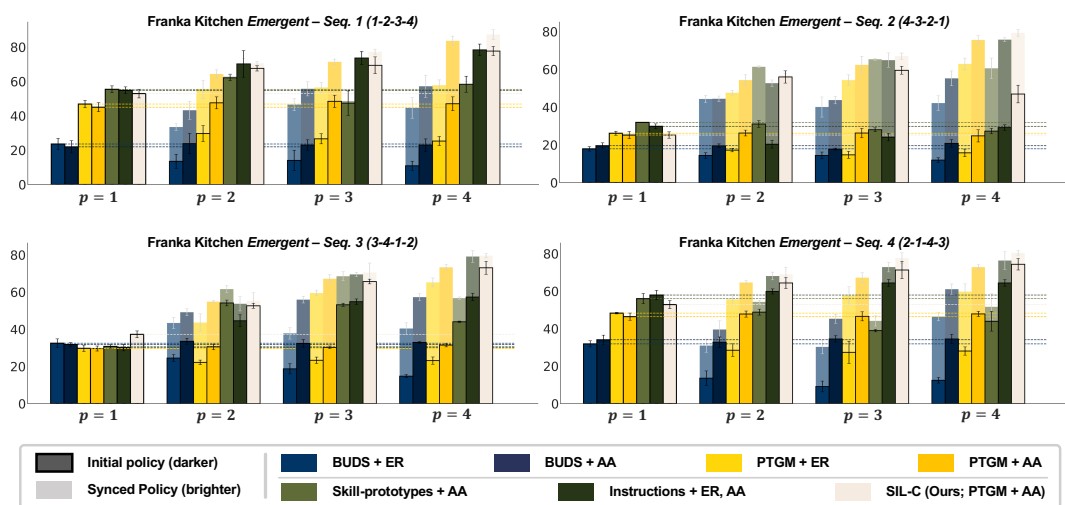

Figure 9: Phase-wise normalized rewards across four Kitchen *Emergent* Skill Incremental datastreams, each corresponding to a different permutation of skill execution order. Each bar group shows performance for different methods under both initial and retrained (Synced) policies.

(*BwSC*), which captures the evolution of the initial policy over phases, and forward skill compatibility (*FwSC*), measured after the skill policies have been resynchronized and retrained. In the *BwSC* setting, SIL-C consistently achieves the highest normalized rewards or matches the performance of methods that use pre-defined semantic skills. After retraining(Synced Policy) in the *FwSC* setting, SIL-C continues to show the highest performance across all phases.

Each sequence corresponds to a permutation of phase orders from Table 6. In **Seq. 2**, the dataset of first phase does not include demonstrations for `open microwave`, so even methods that rely on pre-defined semantic skills fail to execute this task in early phases. Once policies are retrained, SIL-C, SIL with semantic skills, and PTGM+AA reach similar levels of performance. A similar issue appears in **Seq. 3**, where demonstrations for `turn on top burner` are absent in the initial phase. Only SIL-C maintains high performance throughout, showing strong skill-policy compatibility despite the delayed exposure to this skill. **Seq. 4** follows the same pattern for `open slide cabinet`, where again only SIL-C maintains high rewards both before and after retraining.

Additionally, methods that combine semantic retrieval with skill prototypes exhibit decreasing performance over successive phases, particularly in the *Emergent* skill incremental setting. As the diversity of skills increases, these methods are more susceptible to incorrect skill retrieval, which in some cases results in performance degradation relative to earlier phases. Overall, SIL-C is the only method that consistently maintains skill-policy compatibility without requiring prior knowledge of future skills. This property is essential for scalable, SIL.

Table 13: Additional experiments for testing robustness to incremental ordering. Experiments were conducted on the Franka Kitchen environment using various combinations of phase orderings. Each row represents a baseline method, organized according to the skill interface setting, hierarchical agent architecture, and the specific SIL algorithm applied for skill decoder updates. The * symbol indicates methods requiring pre-specified semantic skill labels during both policy and decoder training phases. The left columns present the *BwSC* results, the right columns show the *FwSC* results, while the central columns report the overall combined performance. The best results are indicated in **bold**. In this table, SIL-C utilizes the Type I (PTGM + AA) approach as its foundational architecture.

**Kitchen : *Emergent SIL* Seq. 1 (1-2-3-4)**

| | | | *Initial* | *BwSC* (*Initial*) | | Overall performance | | *FwSC* (*Synced*) | | *Final* |
|---|---|---|---|---|---|---|---|---|---|---|
| Type | Skill Interface | SIL Algo. | FWT (%) | BWT (%) | AUC (%) | BWT (%) | AUC (%) | BWT (%) | AUC (%) | FWT (%) |
| I | Skill Segments [62, 10] (BUDS) | ER | 23.5$_{\pm3.8}$ | -10.8$_{\pm8.1}$ | 15.5$_{\pm2.3}$ | 3.6$_{\pm7.0}$ | 26.6$_{\pm2.8}$ | 17.9$_{\pm6.5}$ | 37.0$_{\pm3.0}$ | 44.5$_{\pm7.0}$ |
| | | AA | 21.9$_{\pm4.1}$ | 1.3$_{\pm4.4}$ | 22.9$_{\pm4.0}$ | 15.7$_{\pm3.5}$ | 35.3$_{\pm2.9}$ | 30.1$_{\pm3.9}$ | 44.4$_{\pm2.6}$ | 57.1$_{\pm7.5}$ |
| | Subgoal Bins [60] (PTGM) | ER | 46.8$_{\pm2.5}$ | -19.7$_{\pm4.4}$ | 32.1$_{\pm2.9}$ | -5.0$_{\pm2.8}$ | 42.5$_{\pm1.4}$ | 9.7$_{\pm3.3}$ | 54.0$_{\pm1.8}$ | 57.7$_{\pm3.6}$ |
| | | AA | 45.0$_{\pm3.1}$ | 2.6$_{\pm0.9}$ | 46.9$_{\pm3.7}$ | 15.4$_{\pm1.7}$ | 58.2$_{\pm1.7}$ | 28.1$_{\pm4.1}$ | 66.1$_{\pm0.6}$ | 83.6$_{\pm3.1}$ |
| II | Skill-prototypes [11]* | AA | 55.1$_{\pm1.8}$ | 0.5$_{\pm3.1}$ | 55.5$_{\pm2.7}$ | 0.5$_{\pm2.8}$ | 55.7$_{\pm2.7}$ | 0.9$_{\pm2.2}$ | 55.8$_{\pm2.4}$ | 56.9$_{\pm4.9}$ |
| | Instructions [12]* | ER | 54.0$_{\pm2.4}$ | 18.5$_{\pm4.0}$ | **67.8$_{\pm1.1}$** | 19.1$_{\pm2.7}$ | 70.3$_{\pm1.0}$ | 19.7$_{\pm2.0}$ | 68.7$_{\pm1.9}$ | 77.4$_{\pm2.8}$ |
| III | SIL-C | AA | 52.9$_{\pm2.9}$ | 18.6$_{\pm1.2}$ | 66.8$_{\pm2.6}$ | 22.0$_{\pm2.4}$ | **71.8$_{\pm1.3}$** | 25.4$_{\pm4.0}$ | **71.9$_{\pm0.9}$** | 87.2$_{\pm3.2}$ |
| - | Subgoal Bins [60] | Joint | - | - | - | - | - | - | - | 86.9$_{\pm2.2}$ |

**Kitchen : *Emergent SIL* Seq. 2 (4-3-2-1)**

| | | | *Initial* | *BwSC* (*Initial*) | | Overall performance | | *FwSC* (*Synced*) | | *Final* |
|---|---|---|---|---|---|---|---|---|---|---|
| Type | Skill Interface | SIL Algo. | FWT (%) | BWT (%) | AUC (%) | BWT (%) | AUC (%) | BWT (%) | AUC (%) | FWT (%) |
| I | Skill Segments [62, 10] (BUDS) | ER | 17.9$_{\pm1.3}$ | -4.4$_{\pm2.0}$ | 14.6$_{\pm1.1}$ | 9.9$_{\pm2.3}$ | 26.4$_{\pm1.7}$ | 24.2$_{\pm2.8}$ | 36.1$_{\pm1.9}$ | 42.1$_{\pm4.5}$ |
| | | AA | 19.5$_{\pm1.7}$ | -0.2$_{\pm2.1}$ | 19.4$_{\pm0.9}$ | 14.0$_{\pm2.2}$ | 31.5$_{\pm1.3}$ | 28.2$_{\pm2.4}$ | 40.7$_{\pm1.7}$ | 55.2$_{\pm4.5}$ |
| | Subgoal Bins [60] (PTGM) | ER | 26.1$_{\pm1.2}$ | -10.3$_{\pm2.5}$ | 18.4$_{\pm1.2}$ | 9.3$_{\pm1.3}$ | 34.1$_{\pm0.3}$ | 28.8$_{\pm1.1}$ | 47.7$_{\pm1.0}$ | 62.8$_{\pm3.5}$ |
| | | AA | 25.2$_{\pm2.2}$ | 0.6$_{\pm1.7}$ | 25.6$_{\pm2.5}$ | 19.7$_{\pm1.0}$ | 42.1$_{\pm1.6}$ | 38.9$_{\pm2.5}$ | 54.3$_{\pm1.8}$ | 75.5$_{\pm2.7}$ |
| II | Skill-prototypes [11]* | AA | 31.9$_{\pm0.2}$ | -3.1$_{\pm0.2}$ | 29.5$_{\pm0.3}$ | 13.7$_{\pm1.1}$ | 43.6$_{\pm1.1}$ | 30.5$_{\pm2.0}$ | 54.7$_{\pm1.6}$ | 60.5$_{\pm6.3}$ |
| | Instructions [12]* | ER | 29.8$_{\pm1.5}$ | -5.3$_{\pm2.7}$ | 25.8$_{\pm1.2}$ | 14.7$_{\pm2.3}$ | 42.3$_{\pm0.6}$ | 34.6$_{\pm2.5}$ | 55.7$_{\pm0.7}$ | 75.7$_{\pm1.2}$ |
| III | SIL-C | AA | 25.2$_{\pm2.0}$ | 28.9$_{\pm1.6}$ | **46.9$_{\pm3.0}$** | 35.7$_{\pm1.0}$ | **55.8$_{\pm1.8}$** | 42.5$_{\pm2.4}$ | **57.0$_{\pm1.2}$** | 79.3$_{\pm2.0}$ |
| - | Subgoal Bins [60] | Joint | - | - | - | - | - | - | - | 86.9$_{\pm2.2}$ |

**Kitchen : *Emergent SIL* Seq. 3 (3-4-1-2)**

| | | | *Initial* | *BwSC* (*Initial*) | | Overall performance | | *FwSC* (*Synced*) | | *Final* |
|---|---|---|---|---|---|---|---|---|---|---|
| Type | Skill Interface | SIL Algo. | FWT (%) | BWT (%) | AUC (%) | BWT (%) | AUC (%) | BWT (%) | AUC (%) | FWT (%) |
| I | Skill Segments [62, 10] (BUDS) | ER | 32.5$_{\pm2.7}$ | -13.2$_{\pm2.1}$ | 22.6$_{\pm1.3}$ | -2.6$_{\pm2.8}$ | 30.3$_{\pm0.7}$ | 8.0$_{\pm3.5}$ | 38.5$_{\pm0.6}$ | 40.4$_{\pm3.6}$ |
| | | AA | 31.8$_{\pm1.2}$ | 1.2$_{\pm2.0}$ | 32.6$_{\pm0.8}$ | 11.7$_{\pm1.7}$ | 41.8$_{\pm1.2}$ | 22.3$_{\pm1.7}$ | 48.5$_{\pm1.5}$ | 57.2$_{\pm2.2}$ |
| | Subgoal Bins [60] (PTGM) | ER | 29.7$_{\pm2.0}$ | -6.9$_{\pm2.2}$ | 24.5$_{\pm0.8}$ | 9.8$_{\pm3.0}$ | 38.1$_{\pm1.3}$ | 26.4$_{\pm4.3}$ | 49.5$_{\pm2.2}$ | 65.2$_{\pm2.7}$ |
| | | AA | 29.6$_{\pm1.6}$ | 1.1$_{\pm1.5}$ | 30.4$_{\pm0.7}$ | 18.3$_{\pm1.8}$ | 45.3$_{\pm0.2}$ | 35.4$_{\pm2.2}$ | 56.2$_{\pm0.5}$ | 73.3$_{\pm1.6}$ |
| II | Skill-prototypes [11]* | AA | 30.7$_{\pm1.1}$ | 19.7$_{\pm0.5}$ | 45.5$_{\pm0.9}$ | 25.6$_{\pm0.6}$ | 52.7$_{\pm1.1}$ | 31.5$_{\pm0.9}$ | 54.4$_{\pm1.3}$ | 56.7$_{\pm0.8}$ |
| | Instructions [12]* | ER | 30.1$_{\pm2.1}$ | 22.1$_{\pm2.6}$ | 46.7$_{\pm1.3}$ | 29.7$_{\pm2.7}$ | 55.6$_{\pm1.6}$ | 37.3$_{\pm3.2}$ | 58.1$_{\pm2.0}$ | 79.1$_{\pm3.7}$ |
| III | SIL-C | AA | 37.2$_{\pm2.1}$ | 26.5$_{\pm1.6}$ | **57.1$_{\pm1.9}$** | 28.9$_{\pm1.7}$ | **62.0$_{\pm2.4}$** | 31.2$_{\pm2.7}$ | **60.6$_{\pm3.1}$** | 79.5$_{\pm1.3}$ |
| - | Subgoal Bins [60] | Joint | - | - | - | - | - | - | - | 86.9$_{\pm2.2}$ |

**Kitchen : *Emergent SIL* Seq. 4 (2-1-4-3)**

| | | | *Initial* | *BwSC* (*Initial*) | | Overall performance | | *FwSC* (*Synced*) | | *Final* |
|---|---|---|---|---|---|---|---|---|---|---|
| Type | Skill Interface | SIL Algo. | FWT (%) | BWT (%) | AUC (%) | BWT (%) | AUC (%) | BWT (%) | AUC (%) | FWT (%) |
| I | Skill Segments [62, 10] (BUDS) | ER | 31.9$_{\pm1.9}$ | -20.2$_{\pm3.6}$ | 16.7$_{\pm1.0}$ | -8.1$_{\pm3.9}$ | 24.9$_{\pm1.6}$ | 4.0$_{\pm4.7}$ | 34.8$_{\pm2.0}$ | 46.4$_{\pm2.5}$ |
| | | AA | 34.0$_{\pm2.9}$ | -0.1$_{\pm3.4}$ | 33.9$_{\pm2.1}$ | 7.2$_{\pm3.2}$ | 40.2$_{\pm1.6}$ | 14.6$_{\pm3.2}$ | 44.9$_{\pm1.1}$ | 61.0$_{\pm3.2}$ |
| | Subgoal Bins [60] (PTGM) | ER | 48.3$_{\pm0.5}$ | -20.3$_{\pm4.0}$ | 33.0$_{\pm2.8}$ | -5.5$_{\pm2.5}$ | 43.6$_{\pm2.2}$ | 9.4$_{\pm2.4}$ | 55.3$_{\pm2.2}$ | 59.7$_{\pm4.8}$ |
| | | AA | 46.4$_{\pm2.2}$ | 1.0$_{\pm2.0}$ | 47.1$_{\pm1.9}$ | 11.4$_{\pm2.1}$ | 56.2$_{\pm0.9}$ | 21.8$_{\pm2.7}$ | 62.7$_{\pm0.3}$ | 72.8$_{\pm1.4}$ |
| II | Skill-prototypes [11]* | AA | 56.0$_{\pm3.2}$ | -12.1$_{\pm4.5}$ | 46.9$_{\pm1.4}$ | -9.1$_{\pm5.3}$ | 48.2$_{\pm2.0}$ | -6.0$_{\pm6.2}$ | 51.5$_{\pm2.2}$ | 51.7$_{\pm8.0}$ |
| | Instructions [12]* | ER | 57.9$_{\pm2.9}$ | 4.9$_{\pm2.2}$ | 61.6$_{\pm1.6}$ | 9.7$_{\pm2.8}$ | 66.2$_{\pm1.1}$ | 14.5$_{\pm3.3}$ | 68.8$_{\pm1.0}$ | 76.3$_{\pm5.4}$ |
| III | SIL-C | AA | 52.8$_{\pm2.4}$ | 17.2$_{\pm1.0}$ | **65.7$_{\pm3.0}$** | 20.0$_{\pm0.9}$ | **69.9$_{\pm2.3}$** | 22.9$_{\pm2.8}$ | **69.9$_{\pm2.4}$** | 80.4$_{\pm1.4}$ |
| - | Subgoal Bins [60] | Joint | - | - | - | - | - | - | - | 86.9$_{\pm2.2}$ |

# D  Additional Analysis

## D.1  Extended Skill and Subtask Space Resolution Analysis

Table 14 reports the full results corresponding to Figure 6, showing the performance of SIL-C (with PTGM and AA) under different skill and subtask space resolutions. Following the setup in Section 6.3, we test both halved and doubled values of $|\bar{\mathcal{Z}}_p|$ and $|\mathcal{G}_\tau|$ relative to the default configuration.

Table 14: Performance under varying skill and subtask clustering configurations in Kitchen *Emergent SIL* scenario. All baselines use the same SIL algorithm AA for the decoder. Analysis with varying $|\mathcal{G}_\tau|$ and $|\bar{\mathcal{Z}}_p|$ in Kitchen *Emergent SIL* scenario. Here, $|\mathcal{G}_\tau|$ denotes the number of subtasks into which expert demonstration data $\mathcal{D}_\tau$ are segmented and $|\bar{\mathcal{Z}}_p|$ indicates the number of skills derived from the datastream $\mathcal{D}_p$ at each phase $p$.

| Kitchen : *Emergent SIL* | | | | | | | | | | | |
|---|---|---|---|---|---|---|---|---|---|---|---|
| Baselines | | | | *Initial* | *BwSC* (*Initial*) | | Overall performance | | *FwSC* (*Synced*) | | *Final* |
| Type | Skill Interface | $|\bar{\mathcal{Z}}_p|$ | $|\mathcal{G}_\tau|$ | FWT (%) | BWT (%) | AUC (%) | BWT (%) | AUC (%) | BWT (%) | AUC (%) | FWT (%) |
| I | Sub-goal Bins [60] (PTGM) | 10 | - | 46.7±2.0 | -0.2±2.1 | 46.2±1.8 | 10.5±1.4 | 55.3±2.6 | 21.1±1.7 | 62.2±3.5 | 76.0±4.9 |
| | | 20 | - | 45.0±3.1 | 2.6±0.9 | 46.9±3.7 | 15.4±1.7 | 58.2±1.7 | 28.1±4.1 | 66.1±0.6 | 83.6±3.1 |
| | | 40 | - | 52.8±1.6 | -0.5±0.2 | 52.4±1.5 | 12.9±0.9 | 63.8±1.2 | 26.2±1.7 | 72.4±1.2 | 86.4±5.4 |
| III | SIL-C (w/ PTGM) | 10 | 10 | 51.6±2.1 | 3.9±2.5 | 54.5±0.9 | 11.6±1.6 | 61.5±1.0 | 19.4±1.7 | 66.1±2.0 | 76.8±2.1 |
| | | | 20 | 49.8±2.2 | 5.0±2.3 | 53.5±3.0 | 14.0±2.5 | 61.8±2.2 | 23.0±3.5 | 67.1±1.6 | 78.1±2.6 |
| | | | 40 | 51.7±2.6 | 4.2±3.5 | 54.9±3.6 | 11.3±3.0 | 61.4±3.0 | 18.4±2.7 | 65.5±2.1 | 74.0±3.1 |
| | | 20 | 10 | 48.6±1.7 | 16.0±3.3 | 60.6±1.3 | 27.8±14.1 | 67.1±0.7 | 27.1±0.7 | 68.9±1.2 | 81.3±1.8 |
| | | | 20 | 52.9±2.9 | 18.6±1.2 | 66.8±2.6 | 22.0±2.4 | 71.8±1.3 | 25.4±4.0 | 71.9±0.9 | 87.2±3.2 |
| | | | 40 | 52.3±1.6 | 16.1±1.2 | 64.6±1.6 | 20.8±0.7 | 70.4±1.5 | 25.6±0.5 | 71.7±1.6 | 86.2±3.1 |
| | | 40 | 10 | 55.1±1.5 | 18.8±2.3 | 69.2±1.0 | 21.5±1.7 | 73.8±0.4 | 24.2±2.1 | 73.3±1.6 | 88.0±2.4 |
| | | | 20 | 54.0±1.2 | 18.8±2.2 | 68.1±0.5 | 22.5±1.6 | 73.3±0.7 | 26.1±1.7 | 73.6±1.5 | 88.8±3.4 |
| | | | 40 | 55.5±1.0 | 19.7±2.2 | 70.3±1.8 | 23.4±1.4 | 75.5±1.3 | 27.1±1.4 | 75.8±1.1 | 90.5±0.2 |

First, we analyze Type I (Subgoal Bins). At $|\bar{\mathcal{Z}}_p| = 10$, the skill space resolution is too coarse, leading to limited overall performance (55.3% AUC). The *BwSC* evaluation group shows weak results (46.2%), and the BWT score remains negative. Increasing to $|\bar{\mathcal{Z}}_p| = 20$ improves overall AUC to 58.2%, and *FwSC* evaluation results rise to 66.1%, but BWT still indicates poor transfer. At $|\bar{\mathcal{Z}}_p| = 40$, Type I gains further, with 63.8% AUC and 72.4% in *FwSC* evaluation, yet *BwSC* remains low (52.4%). These results show that while higher skill space resolution helps Type I, static subgoal binning as the interface is unable to leverage newly added skills without retraining the policy, which is reflected by persistently low BWT in the *BwSC* evaluation group.

When $|\bar{\mathcal{Z}}_p| = 10$, Type III shows clear improvements over Type I. It achieves higher Initial FWT (51.6–51.7%) and stronger AUC in the *BwSC* evaluation group (up to 54.9%). Importantly, the BWT score is positive, indicating that SIL-C enables backward transfer even at coarse resolution, which is not observed in Type I.

When $|\bar{\mathcal{Z}}_p| = 20$, Type III shows consistent improvements over Type I. It achieves higher Initial FWT (48.6–52.3%) and stronger results in both the *BwSC* (up to 68.6%) and *FwSC* (up to 71.9%) evaluation groups. Importantly, the BWT score increases significantly, confirming that SIL-C leverages moderate resolution more effectively than static binning.

When $|\bar{\mathcal{Z}}_p| = 40$, Type III achieves the strongest improvements. It achieves higher Initial FWT (up to 55.5%) and consistently high results in both the *BwSC* (70.3%) and *FwSC* (75.8%) evaluation groups. Most notably, the BWT score peaks at 19.7, the best among all configurations. In contrast, Type I also benefits from higher resolution but remains far behind, with 63.8% overall AUC, 72.4% in *FwSC*, 52.4% in *BwSC*, and negative BWT.

In summary, increasing skill resolution improves both overall performance (AUC) and evaluation results across *BwSC* and *FwSC* groups. Moreover, SIL-C (Type III) consistently leverages this advantage to achieve strong skill-policy compatibility, which is not observed in the static subgoal binning baseline (Type I).

## D.2  Subtask Space Analysis on Few-shot Imitation Learning

Table 15 expands on Table 2 by replacing the prototype-based representation in the SIL-C subtask space with an element-wise retrieval strategy across 1-5 shot scenarios. Element-wise retrieval consistently yields stronger performance than prototype-based representations.

In the 5-shot setting, for example, it achieves a +4.2% absolute gain in overall AUC (66.6% vs. 62.9%) and +3.3% in final FWT (79.1% vs. 75.8%). This performance gap persists across shot counts, highlighting the benefit of fine-grained memory when the budget allows. Notably, SIL-C supports both representations and can incorporate element-wise memory when additional storage is available.

Table 15: Comparison between Gaussian prototypes (*Prototype*) and simple instance accumulation (*Element*) in the subtask space of the Franka Kitchen *Emergent* skill incremental scenario under varying shot settings (1, 3, 5). Memory per task indicates the number of embedding vectors stored in the subtask space for a single task (e.g., $\mu$ or $\Sigma$), with relative usage at 100% corresponding to the configuration in Table 1.

**Kitchen : *Emergent SIL***

| Ablations | | | *Initial* | *BwSC (Initial)* | | Overall performance | | *FwSC (Synced)* | | *Final* |
|---|---|---|---|---|---|---|---|---|---|---|
| Instance | Shots | Memory per Task | FWT (%) | BWT (%) | AUC (%) | BWT (%) | AUC (%) | BWT (%) | AUC (%) | FWT (%) |
| Prototype | 5 | 160 (100%) | 47.4±3.2 | 11.3±2.4 | 55.9±4.6 | 17.5±1.2 | 62.4±2.7 | 23.6±4.2 | 65.1±2.3 | 75.8±4.4 |
| | 3 | 160 (100%) | 44.4±1.8 | 11.2±2.5 | 52.8±2.9 | 14.9±2.2 | 57.2±1.8 | 18.6±3.3 | 58.4±1.2 | 68.5±4.0 |
| | 1 | 160 (100%) | 43.1±5.4 | 11.9±2.8 | 52.0±5.3 | 15.7±3.1 | 56.5±3.7 | 19.4±4.8 | 57.6±2.9 | 65.7±3.6 |
| Element-wise | 5 | 1000 (625%) | 50.3±4.6 | 13.9±4.3 | 60.7±6.4 | 20.6±1.6 | 66.6±4.4 | 24.3±5.3 | 68.5±2.5 | 79.1±3.4 |
| | 3 | 600 (375%) | 48.9±3.0 | 15.6±4.6 | 60.6±2.5 | 19.3±4.2 | 65.4±2.1 | 23.0±3.9 | 66.1±1.3 | 77.3±3.6 |
| | 1 | 200 (125%) | 42.3±3.6 | 8.3±3.3 | 48.5±4.3 | 14.6±2.4 | 54.8±2.6 | 21.0±3.8 | 58.0±1.9 | 64.6±3.2 |

## D.3 Robustness to Observation Noise

We extend the analysis in Table 3, which compares SIL-C against PTGM+AA across different noise levels with Gaussian noise scaled by factors of ×1 to ×5 in Franka Kitchen [65].

**SIL-C maintains compatibility under noise.** As shown in Table 3, SIL-C consistently outperforms PTGM+AA across all noise levels and evaluation metrics. Most notably, SIL-C maintains positive backward transfer (BWT) even under extreme noise (×5: 4.3% vs. -1.2%), while PTGM+AA suffers from catastrophic forgetting with negative BWT. This gap becomes more pronounced in the final phase with approximately 80 skills, where SIL-C achieves 23.1% FWT compared to 15.9% for PTGM+AA under ×5 noise.

**Trajectory matching provides noise resilience.** The robustness gap stems from fundamental architectural differences. Under noise, high-level policies produce increasingly unreliable subtask selections. The bilateral lazy learning interface in SIL-C handles this through: (i) skill validation that detects out-of-distribution subtasks, and (ii) skill hooking that remaps to appropriate skills based on trajectory similarity. The static skill mapping of PTGM+AA, however, cannot adapt to noise-induced distributional shifts, leading to compounding errors in long-horizon tasks.

**Performance scales with skill repertoire size.** The advantage of SIL-C grows as training progresses. Under ×3 noise, the performance gap increases from 0% (Initial FWT: 35.6% vs. 35.8%) to 12.4% (Final FWT: 52.9% vs. 40.5%). This scaling effect suggests that skill validation and hooking become more valuable as the skill library expands, opening the possibility for more robust lifelong learning systems that can maintain performance even in challenging real-world like conditions.

## D.4 Analysis on Distance Metrics for Instance-based Classification

Table 16: Comparison across distance metrics (Mahalanobis, Euclidean) and sub-cluster selection strategies (Fixed, Auto) in the Franka Kitchen Emergent Scenario. We define 100% as the configuration matching the number and memory usage of sub-clusters per prototype used in Table 1. *Fixed* uses a predetermined number of sub-clusters, while *Auto* determines the number of sub-clusters using the silhouette score. The left columns report initial forward transfer (*Initial*) and backward skill compatibility (*BwSC*), which evaluate existing policies with updated skills. The right columns report forward skill compatibility (*FwSC*), assessing the effectiveness of new policies with learned skills, and final forward transfer (*Final*). The central columns indicate overall performance, combining *BwSC* and *FwSC*.

**Kitchen : *Emergent SIL***

| Baselines | | | *Initial* | *BwSC (Initial)* | | Overall performance | | *FwSC (Synced)* | | *Final* |
|---|---|---|---|---|---|---|---|---|---|---|
| Type | Distance Type | Sub-clusters per prototype | FWT (%) | BWT (%) | AUC (%) | BWT (%) | AUC (%) | BWT (%) | AUC (%) | FWT (%) |
| III | Mahalanobis | Fixed $K_{\bar{z}}, K_{\bar{g}} = 4.0$ (100%) | 52.9±2.9 | 18.6±1.2 | 66.8±2.6 | 22.0±2.4 | 71.8±1.3 | 25.4±4.0 | 71.9±0.9 | 87.2±3.2 |
| | | Auto $K_{\bar{z}}, K_{\bar{g}} \approx 2.5$ (63.5%) | 52.7±3.0 | 16.6±2.5 | 65.1±2.0 | 19.6±2.6 | 69.5±1.6 | 22.7±2.9 | 69.7±1.4 | 80.6±3.2 |
| | Euclidean | Fixed $K_{\bar{z}}, K_{\bar{g}} = 4.0$ (50%) | 52.2±4.7 | 5.5±2.3 | 56.3±3.1 | 14.3±4.7 | 64.5±0.8 | 23.2±7.2 | 69.6±0.7 | 81.0±3.6 |
| | | Fixed $K_{\bar{z}}, K_{\bar{g}} = 8.0$ (100%) | 54.0±2.6 | 5.6±2.8 | 58.2±1.8 | 14.4±1.9 | 66.3±1.7 | 23.1±1.4 | 71.3±2.0 | 82.9±2.8 |

We ablate the choice of distance metric in the instance-based classifier, comparing Mahalanobis and Euclidean distances with varying memory configurations as shown in Table 16. Memory usage

is measured relative to storing 4 sub-clusters with both mean and covariance per prototype (100% baseline).

**Mahalanobis distance enables better trajectory matching.** Across all configurations, Mahalanobis distance consistently outperforms Euclidean distance. With the baseline configuration, Mahalanobis achieves 66.8% AUC in BwSC while Euclidean only reaches 56.3%, a 10.5% gap. This difference is most pronounced in backward compatibility, where Mahalanobis maintains 18.6% BWT compared to 5.5% for Euclidean, demonstrating superior ability to align evolving skill distributions with existing policies.

**Memory-performance trade-off with automatic clustering.** When using automatic sub-cluster selection via silhouette score, the Mahalanobis variant reduces memory to 63.5% while maintaining competitive performance (65.1% vs. 66.8% AUC in BwSC). This 3.2% performance drop for 36.5% memory savings presents a viable option for resource-constrained deployments, as the SIL-C still significantly outperforms Euclidean variants even with reduced memory.

**Richer skill distribution modeling improves compatibility.** The performance gap between distance metrics demonstrates the importance of capturing skill distributions accurately. Euclidean distance with 4 sub-clusters (50% memory) achieves only 56.3% AUC in BwSC, and even doubling to 8 sub-clusters (100% memory) reaches just 58.2% AUC. In contrast, Mahalanobis distance with 4 sub-clusters achieves 66.8% AUC despite using diagonal covariance for efficiency. This 8.6%-10.5% gap shows that modeling variance, even in simplified diagonal form, is more effective than simply adding more isotropic prototypes. The Euclidean approach scales poorly because increasing prototypes cannot compensate for the lack of directional information, while Mahalanobis captures essential trajectory variations that distinguish functionally different skills. This enables more accurate skill validation and hooking, particularly critical for backward compatibility where precise distribution matching determines whether existing policies can leverage new skills.

### D.5    Analysis on Instance-based Classifier Thresholding

Figure 10 shows the stability of confidence intervals used for threshold selection in the *Emergent SIL* scenario. Overall, varying the threshold between 80% and 99% results in minimal differences in both the confidence intervals and performance. This indicates that our bounding method does not simply adjust the threshold for skill validation, but instead serves as an effective mechanism for filtering out out-of-distribution cases.

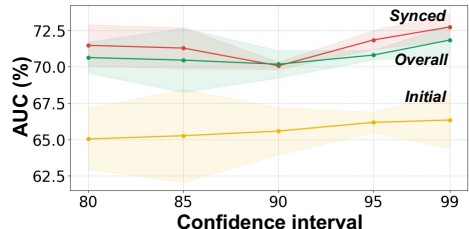

Figure 10: Ablation on SIL-C threshold

### D.6    Memory and Time Analysis

**Memory.** Each instance-based classifier stores Gaussian prototypes in both skill and subtask spaces. The memory cost for skill prototypes is $dim \times |\mathcal{Z}_p| \times K_z \times 2$ float values, and for subtask prototypes it is $dim \times |\mathcal{G}_\tau| \times K_g \times 2$. Here, $dim$ is the dimension of the state or subgoal embedding. Each $(\mu, \Sigma)$ pair is stored using a diagonal covariance matrix. $K_z$ and $K_g$ denote the average number of sub-clusters per skill and subtask, respectively.

In the Kitchen *Emergent* and *Explicit* skill incremental settings with $dim = 60$, skill prototypes occupy 150KB and subtask prototypes occupy 37.5KB, as shown in Table 1. In the Meta-World benchmark with $dim = 140$, the memory usage increases to 350KB for skill prototypes and 87.5KB for subtask prototypes.

**Time.** Each Mahalanobis distance computation with diagonal covariance requires $4 \times dim$ floating point operations: $dim$ subtractions, $dim$ squarings, $dim$ divisions, $dim - 1$ additions, and one square root. For $dim = 60$, the total is 240 FLOPs per $(\mu, \Sigma)$ pair.

In the best case, a proposed $z_h$ is directly accepted as the skill $z_l$ during the skill validation process using a selected subgoal from the subtask space. This process requires $(|\mathcal{G}_\tau| \times K_g + 1) \times 240$ FLOPs. In the worst case, $z_h$ is rejected during skill validation, and skill hooking requires selecting candidate skills $\mathcal{Z}'$ and scoring them to choose the appropriate skill $z_l$. The total computational cost is up to $\{(|\mathcal{G}_\tau| \times K_g + 1) + (2 \times |\mathcal{Z}_p| \times K_z + 1)\} \times 240$ FLOPs.

Table 17: Mean and variance (ms) of evaluation function timing in the Kitchen *Emergent* Skill Incremental scenario. *kbls* denotes the task with the following sequence: `move kettle, turn on bottom burner, light switch, open slide cabinet`, and *btls* denotes: `turn on bottom burner, turn on top burner, light switch, open slide cabinet`.

| Baselines | | *kbls-Initial* | | *btls-Initial* | | *kbls-Final* | | *btls-Final* | |
|---|---|---|---|---|---|---|---|---|---|
| Type | Method | Mean | Var | Mean | Var | Mean | Var | Mean | Var |
| I | PTGM + AA | 28.089 | 2.042 | 28.117 | 1.784 | 27.693 | 1.978 | 28.052 | 2.191 |
| III | SIL-C + (PTGM + AA) | 29.042 | 2.341 | 27.782 | 2.324 | 28.787 | 2.732 | 28.538 | 2.320 |

Table 17 reports the inference time (mean and variance) of the hierarchical model, consisting of the policy, interface, and skill decoder, in the Kitchen *Emergent* Skill Incremental scenario. All timings were measured on an AMD Ryzen 9 7950X3D CPU with a single NVIDIA RTX 4090 GPU, running Ubuntu 22.04, CUDA 12.4, and driver version 550.144.03. The *Initial* phase ($p = 1$) corresponds to the evaluation after 20 skill prototypes were generated, while the *Final* phase ($p = 4$) reflects evaluation after the final phase with a total of 80 skills. Compared to Type I, the mean inference time of Type III increased by 0.953 ms and 1.094 ms in the *kbls-Initial* and *kbls-Final*, respectively. In the *btls-Initial*, the mean time decreased by 0.335 ms, while in the *btls-Final*, it increased by 0.486 ms. While a greater number of skills generally leads to longer retrieval time, the additional overhead was minimal compared to overall policy and skill decoder inference time, and variation due to runtime system state was more prominent in practice.

