# OpenReview forum: "Policy Compatible Skill Incremental Learning via Lazy Learning Interface"
_NeurIPS.cc/2025/Conference — NeurIPS 2025 spotlight_

### Official Review · Reviewer_JDZP · 2025-06-13

**Clarity:** 3
**Significance:** 3
**Originality:** 3
**Rating:** 4
**Confidence:** 3

**Summary:**

This paper addresses the problem of Skill Incremental Learning, where an agent incrementally expands and refines its skill set over time through interaction or the addition of new data. The focus is on a particular challenge in this setting: skill compatibility—specifically within hierarchical architectures where the skill module and policy module are updated at different rates, leading to potential inconsistencies. The authors propose two mechanisms to maintain compatibility:
- BwSC Ensures that newly updated skills remain usable by previously trained policies.
- FwSC Allows previously trained skills and policies to utilize newly added skills effectively.

**Questions:**

- Clarify the motivation behind the assumed architectural setup. Why is a hierarchical skill-policy architecture necessary, and why are updates asynchronous? Is this common in practice or a design choice unique to this work?
- Add evaluation or discussion on how the method would handle noisy inputs, domain shifts, or other real-world challenges. Even a small-scale real-world test or robustness analysis would significantly strengthen the contribution.

**Ethical Concerns:**

["NO or VERY MINOR ethics concerns only"]

**Final Justification:**

Thanks, author, for answering my questions and providing additional experiments. I remain positive on this submission; however, I still hope to see more real-world experiments.

**Limitations:**

yes

**Quality:**

3

**Strengths And Weaknesses:**

Strengths
- The paper clearly identifies a practical challenge in skill-based policy architectures and provides a thoughtful solution.
- The proposed compatibility mechanisms are well-motivated and show promising results.
- Extensive experiments in simulated environments are used to validate the effectiveness of the method.


Weaknesses
- The issue of skill-policy incompatibility appears to arise from the design choice of using a hierarchical architecture with asynchronous updates. However, the paper does not sufficiently motivate this design. Why are skills and policies trained separately or at different rates? Establishing this necessity upfront would better justify the need for compatibility mechanisms.

-  There is no evaluation in more realistic settings, particularly under noisy or imperfect state/action representations. This limits confidence in the generalizability of the approach beyond clean simulations.

---

> ### Author Rebuttal · Authors · 2025-07-30
>
> We thank the reviewer for their thoughtful and constructive feedback. We deeply appreciate the time and care taken to evaluate our work, and especially value the insightful suggestions that highlight both the strengths of our contribution and areas for refinement.
>
> ---
> > W1. Q1. Clarify the motivation behind the assumed architectural setup. Why is a hierarchical skill-policy architecture necessary, and why are updates asynchronous? Is this common in practice or a design choice unique to this work?
>
> Here, we clarify the motivation behind the Policy Compatible Skill Incremental Learning scenario, which forms the basis of our architectural setup.
> * Embodied agents for complex decision-making (hierarchical skill-policy architecture): (Section 2, paragraph 1) [1,2,3]
>   * Hierarchical architectures are widely adopted to address complex, long-horizon tasks (such as those in the Franka Kitchen environment) by decomposing them into manageable subtasks.
>   * This decomposition facilitates more tractable decision-making and enables the modular reuse of pre-trained skills, which is a standard practice in robotic learning.
>   * By decoupling high-level decision-making from low-level control, the hierarchical structure improves learning efficiency and execution flexibility across diverse tasks.
>
> * Lifelong learning for modular architecture deployment configurations. (asynchronous updates): (Section 2, paragraph 2) [4,5]
>   * Asynchronous updates are common in lifelong learning scenarios where modularized components are trained on distinct data sources with different objectives. For example, in producer-consumer deployment configurations, the producer pre-trains and deploys a general-knowledge model, while the consumer uses or customizes it for specific tasks.
>   * Differing update frequencies naturally arise across modules due to variations in how frequently their respective data sources are updated. This makes it essential to maintain compatibility between evolving skills and policies.
>   * Modularization thus facilitates asynchronous updates, enabling efficient, task-specific learning without the need to retrain the entire agent system.
>
> To the best of our knowledge, our work is the first to integrate asynchronous modular updates into a hierarchical policy-skill architecture for lifelong robotic learning.
>
> Here, we clarify compatibility problems and solutions for the architectural design process.
> * Forward Skill Compatibility (FwSC)
>   * To ensure FwSC, it is crucial to maintain existing skills without forgetting while allowing for the addition of new ones.
>   * We propose building a common and expandable embedding space for skills and subtasks, using a model expansion paradigm that effectively prevents the problem of explicit forgetting.
>   * Lightweight adapters like LoRA offer an efficient and robust approach to lifelong learning. Rather than modifying existing components, new functionalities are appended modularly.
>   * This strategy is effective in preventing forgetting by structurally isolating newly added knowledge from older skills.
>
> * Backward Skill Compatibility (BwSC)
>   * To ensure BwSC, the policy's high-level decision-making process must be made "lazy". In other words, the agent should defer integrating any newly appended skill until it actually observes a state distribution that makes that skill beneficial, rather than eagerly re‑optimizing the whole hierarchy in advance.
>   * Our core insight is that not every decision-making function needs to be lazy; rather, only specific components should be lazily offloaded when appropriate.
>   * To implement this, each modular model must establish a shared protocol regarding its behavior (e.g., pre-defined skills, subtask labels) and make decisions accordingly.
>   * Specifically, only the "validate subtask and reassign skill" step is lazily offloaded within the shared space (i.e., state space), leveraging a prototype memory to compute trajectory distribution similarity.
>
> Collectively, these design choices enable lifelong learning in complex decision-making agents. We will incorporate this explanation into the introduction to clarify the design choices and their relevance to lifelong embodied agents. We continue the discussion of the connection between the motivation and the implementation in our response to `39nR W2`.
>
> ---
> > W2. Q2. Add evaluation or discussion on how the method would handle noisy inputs, domain shifts, or other real-world challenges. Even a small-scale real-world test or robustness analysis would significantly strengthen the contribution.
>
> To evaluate the robustness of SIL-C to observation noise, we used a model trained on the Kitchen Emergent Skill Incremental scenario in the Franka Kitchen simulator [1], where Gaussian observation noise is present by default as part of the robot's specification. During evaluation, we varied the intensity of this built-in noise by applying different noise scales, specifically scaling the default noise level by factors of ×1, ×2, ×3, and ×5. SIL-C was compared against the Type 1 baseline PTGM+AA under these noise conditions. The results are summarized in the following table.
> |Kitchen Emergent scenario||Initial|BwSC||Overall||FwSC||Final|
> |-|-|-|-|-|-|-|-|-|-|
> |Noise level|Baselines|FWT|BWT|AUC|BWT|AUC|BWT|AUC|FWT|
> |×1|PTGM (AA)|45.0±3.1|2.6±0.9|46.9±3.7|15.4±1.7|58.2±1.7|28.1±4.1|66.1±0.6|83.6±3.1|
> ||SIL-C (AA)|52.9±2.9|18.6±1.2|66.8±2.6|22.0±2.4|71.8±1.3|25.4±4.0|71.9±0.9|87.2±3.2|
> |
> |×2|PTGM (AA)|44.4±4.5|0.8±2.0|45.0±3.4|10.7±2.7|53.5±2.2|20.5±3.9|59.7±1.9|68.6±3.5|
> ||SIL-C (AA)|49.7±0.6|11.3±1.8|58.2±1.5|16.7±1.1|64.0±1.3|22.1±1.9|66.3±1.8|76.4±2.6|
> |
> |×3|PTGM (AA)|35.8±5.3|0.5±3.7|36.1±2.9|3.2±5.6|38.5±1.5|5.9±7.7|40.2±1.7|40.5±4.6|
> ||SIL-C (AA)|35.6±2.2|13.6±4.2|45.8±3.1|15.4±2.9|48.8±1.5|17.3±3.4|48.5±1.0|52.9±3.9|
> |
> |×5|PTGM (AA)|19.7±1.7|-1.2±3.4|18.8±1.7|-1.8±2.4|18.2±0.9|-2.4±2.5|17.9±1.1|15.9±1.6|
> ||SIL-C (AA)|19.6±2.3|4.3±1.3|22.8±2.4|3.8±2.4|22.9±1.4|3.3±4.1|22.1±1.4|23.1±4.1|
>
> Summary of Results:
> * **Overall Performance:**
>   SIL-C consistently outperforms PTGM+AA across all noise levels, demonstrating superior robustness in noisy environments.
> * **Initial Performance:**
>   At the beginning of training (Initial FWT), SIL-C exhibits stronger performance than PTGM+AA. However, under extreme noise conditions (×5), both methods experience a performance drop, and their scores converge and become comparable.
> * **BwSC:**
>   SIL-C maintains a positive Backward Transfer (BWT) across all noise levels, indicating stable retention of previously learned skills. In contrast, PTGM+AA suffers from negative BWT under higher noise, suggesting degradation of earlier knowledge.
> * **FwSC:**
>   SIL-C consistently achieves higher AUC, reflecting better integration of newly learned skills.
> * **Final Performance:**
>   As training progresses and more skills are added (Final FWT), the performance gap between SIL-C and PTGM+AA widens compared to Initial FWT. This divergence is especially pronounced under high-noise conditions.
>
> This robustness becomes particularly important as the agent accumulates a larger repertoire of skills, up to approximately 80 in the final FWT phase. As noise levels increase, SIL-C exhibits a more gradual degradation in performance, reflecting stronger generalization capabilities in noisy environments.
>
> We attribute this robustness to the distinct ways each method handles noisy subtask distributions. In extreme noise conditions, where the high-level policy’s subtask selection becomes less reliable, SIL-C increasingly leverages a prototype memory-based skill validation mechanism, combined with a trajectory distribution similarity-based skill hooking process. These components enable SIL-C to retrieve and deploy skills in a more adaptive and noise-tolerant manner.
>
> In contrast, PTGM+AA relies solely on static subtask identifiers, which fail to account for distributional shift caused by observation noise. As a result, its performance deteriorates more sharply as environmental noise increases. This disparity becomes especially pronounced in long-horizon environments like Franka Kitchen, where even a single misstep can lead to substantial reductions in cumulative reward. Such environments amplify the impact of noise, further underscoring the robustness of SIL-C.
>
> [1] Gupta, Abhishek, et al. "Relay policy learning: Solving long-horizon tasks via imitation and reinforcement learning." CoRL. 2019.
>
> [2] Pertsch, Karl, Youngwoon Lee, and Joseph Lim. "Accelerating reinforcement learning with learned skill priors." CoRL. 2021.
>
> [3] Yuan, Haoqi, et al. "Pre-training goal-based models for sample-efficient reinforcement learning." ICLR. 2024.
>
> [4] Mendez, Jorge A., Harm van Seijen, and Eric Eaton. "Modular lifelong reinforcement learning via neural composition." arXiv preprint. 2022.
>
> [5] Sukhbaatar, Sainbayar, et al. "Branch-train-mix: Mixing expert llms into a mixture-of-experts llm." COLM. 2024.

---

> > ### Author Response · Authors · 2025-08-08
> >
> > Dear JDZP,
> >
> > Thank you again for your thoughtful review of our paper! We prepared our responses to your comments, including clarifications on the motivation behind the architectural setup and a robustness analysis of SIL-C to noisy inputs. As the discussion deadline nears, please let us know if you have any further questions so we can address them in time.

---

### Official Review · Reviewer_UFGb · 2025-07-02

**Clarity:** 1
**Significance:** 2
**Originality:** 3
**Rating:** 4
**Confidence:** 3

**Summary:**

Build up an incremental skill library by applying append-only skill updates and versioning, as inspired by filesystems. This involves a high-level policy passing subgoals to a low-level policy according to a temporal distance estimator, where the low-level policy selects between skills. Each module uses memory as the parameterization, and consists of classifiers that output scores for different selections of skills and goals. The skills are added to by clustering the new data, and described using an energy-based function over the primitive actions. Recovering policies involves selecting the lowest-scoring low-level skill. This method is then evaluated on Franka Kitchen and meta-world environments.

**Questions:**

The lack of clarity for the policy hooking, lazy learning, memory usage, classifiers and other components makes it hard to assess this work properly.

**Ethical Concerns:**

["NO or VERY MINOR ethics concerns only"]

**Final Justification:**

The writing is improved with the author's clarifications

**Limitations:**

See weaknesses

**Paper Formatting Concerns:**

The formatting for citations uses numbering, but I'm not sure if that follows NeurIPS guidelines.

**Quality:**

2

**Strengths And Weaknesses:**

**Strengths:**

The challenge of building up skills without losing current capabilities is fundamental to lifelong learning.

The method draws interesting connections and evokes interesting ideas related to saving skills.

The evaluation environments are challenging

**Weaknesses:**

There appears to be a typo in the intro line 48.

While the method is a fairly sizable system, Figure 2 does not clearly articulate the core premise of the system, that is, what data it expects to receive and how it should transform that data into an appropriate set of skills. While this may be contained in the figure, it is obfuscated by the layers of complexity.

Either the problem statement for Figure 2 is inadequate for describing how certain components are being used, especially how the memory system for both states and skills works, and what the inputs and outputs of the modules are. While it is possible to make assumptions from other work, it is not clear because of the way these components are expressed.
It is also not clear what is done if a query-label pair is rejected (line 140). This is because the concept of skill hooking has not been introduced yet.

Throughout the paper, the ordering is confusing, since many of the terms used are not clearly explained, nor are they common practice. For example, lazy learning and skill hooking are terms used but not explained, even intuitively, let alone formally, and without any reference to when they will be described in detail.

Several key issues with adding skills incrementally are not directly addressed: What happens to the skill space when there are too many skills, so that clustering and classification become infeasible? How do the classifiers both scale and handle errors? Is this method able to improve existing skills or transfer them, since this is another one of the core promises of incremental skill learning, and if not, is this a core issue with the philosophy of keeping old data? How does this method apply to settings where some of the data may be suboptimal?
The description of lower score being better is confusing because score is typically higher is better.

In like 182, it does not make sense to introduce new skills by training the skill decoder, since the skills are introduced by modifying the environment. There must be some sequence of skills to describe when the values will come out.

As far as I am aware, the baseline methods are much simpler. It seems important to guarantee that the core principles behind SIL-C are the source of its advantage, and not just the presence of more hyperparameter tuning. However, there are no clear descriptions of what the core principles of this method might be, and the ablative experiments removing different components are terse and do not provide much insight into how the results imply that the component adds benefit. For policy hooking, this is because it is not clear why random should do better than a policy, and varying the classifier does not help suggest why the classifier itself is useful.

---

> ### Author Rebuttal · Authors · 2025-07-30
>
> We thank the reviewer for the time and effort invested in carefully reading not only the main manuscript but also the appendix. We truly appreciate the detailed and constructive feedback, which clearly identifies points for improvement and gives us the opportunity to address them. The reviewer’s thoughtful comments on clarity and presentation are especially valuable in enhancing the overall quality of the paper, and we have carefully reflected on these suggestions with a strong commitment to improving the paper.
>
> ---
> > Core claims and motivating experiment of SIL-C
>
> To enable lifelong learning in a modularized agent with a hierarchical architecture capable of complex decision making, we identify the following key challenges:
> * **Forward Skill Compatibility (FwSC)**: ensures that future downstream policies can effectively utilize newly added skills.
> * **Backward Skill Compatibility (BwSC)**: ensures that existing downstream policies can continue to use newly added skills **without requiring re-training of each module** and, when applicable, benefit from improved policy performance.
>
> Together, these two principles not only aim to preserve the agent’s existing capabilities during Skill Incremental Learning (SIL) but also **enhance the performance of existing policies** by enabling the effective integration of newly added skills.
>
> To motivate our problem setting, we highlight that even multi-task learning, often considered an oracle in lifelong learning, fails to resolve this issue through the extended Kitchen Emergent SIL scenario results in Table 1.
> |Baselines|||Initial|BwSC||Overall||FwSC||Final|
> |-|-|-|-|-|-|-|-|-|-|-|
> |Skill Interface|SIL Algo.|Replay Ratio|FWT|BWT|AUC|BWT|AUC|BWT|AUC|FWT|
> |PTGM|ER|10%|46.8±2.5|-19.7±4.4|32.1±2.9|-5.0±2.8|42.5±1.4|9.7±3.3|54.0±1.8|57.7±3.6|
> ||ER|50%|48.2±1.9|-9.2±2.2|41.3±2.0|5.8±1.8|53.1±1.8|20.8±1.9|63.8±1.7|74.7±2.2|
> ||ER (multi-task)|100%|48.4±1.4|-3.1±1.8|46.1±0.9|11.6±0.9|58.4±1.2|26.2±0.9|68.1±1.7|81.3±4.5|
> |SIL-C (ours)|ER (multi-task)|100%|52.7±2.8|15.2±2.8|64.1±2.6|19.7±2.1|69.6±2.1|24.3±1.7|70.9±1.7|82.7±0.7|
> |
> |PTGM|AA|0%|45.0±3.1|2.6±0.9|46.9±3.7|15.4±1.7|58.2±1.7|28.1±4.1|66.1±0.6|83.6±3.1|
> |SIL-C (ours)|AA|0%|52.9±2.9|18.6±1.2|66.8±2.6|22.0±2.4|71.8±1.3|25.4±4.0|71.9±0.9|87.2±3.2|
>
> We evaluate simple baselines (PTGM with ER or AA) across varying replay ratios (10%, 50%, 100%) to assess FwSC and BwSC. Notably, even multi-task learning, often considered an upper bound of SIL, fails to achieve BwSC, showing near-zero BWT and thus limited reuse of new skills for improving existing policies.
> In contrast, SIL-C with the same configuration as the simple baselines, achieves positive BWT in the BwSC evaluation, indicating that newly acquired skills enhance existing policies without retraining and highlighting SIL-C’s key strength in enabling scalable and compositional lifelong learning through both FwSC and BwSC.
>
> ---
> > W1 - W3. The paper contains some typos, and the explanations in Section 4 and Figure 2 lack clarity.
>
> To improve clarity as suggested, we summarize the key revisions.
> * We correct the percentage point notation and revise the previously misreported value of 18.0 to the correct value of 18.6 in line 48.
> * We will simplify Figure 2.
> * We provide high-level pseudocode in Section 4 for clarity, with details addressed in our response to `39nR W1`. We also clarify and explicitly reference the Appendix A.2 algorithm in the main manuscript.
>
> ---
> > W4. Furthermore, the overall ordering of concepts is difficult to follow.
>
> We acknowledge that certain details may not have been fully elaborated due to space constraints.
> * Revise the introduction to include a brief paragraph that motivates and defines key SIL-C functionalities.
> * We will provide a summary of the architectural motivations, which connects with the discussions presented in our responses to `JDZP W1` and `39nR W2`, thereby offering clearer context prior to the detailed method description in Section 4.
>
> We also provide clarification for key terms used throughout the paper:
> * Skill validation: The process of determining whether a subtask is suitable or effective to be used as a skill under the task context.
> * Skill hooking: A mechanism for intercepting and altering skill outputs at runtime between modules. The term is borrowed from software engineering, where "hooking" refers to dynamically modifying program behavior.
> * Lazy learning [1]: In our work, this refers to an instance-based retrieval approach in which computation is deferred until it is needed.
>
> ---
> > W5. Concerns regarding scalability and skill addition.
>
> Although our primary focus is on achieving skill-policy compatibility, SIL-C offers a robust and easily extensible framework for integrating new skills.
>
> 5-1. Framework scalability
> * SIL-C is reformulated as an instance-based classification problem, and we observe that its performance improves proportionally with classification accuracy (our response on `39nR Q2`).
> * Regarding scalability with respect to the number of skills, the instance-based classification in SIL-C is inherently scalable and can be efficiently implemented using vector database optimization tools such as Faiss [2].
>
> 5-2. Skill improvement and transfer
> * As shown in Figure 4 of Section 5.4, SIL-C can be flexibly applied to various combinations of skill clustering methods and SIL algorithms.
> * In the append-only setting, improving and transferring existing skills essentially involves adding a modified version of the skill while preserving the original, allowing both to coexist.
> * When generating a new skill based on existing ones, the explicit improvement and transfer process is governed by the skill decoder’s learning algorithm. Our method supports two such approaches:
>   * Experience Replay-based approaches, where the quality of a skill is directly and forcibly updated. This is demonstrated in Figure 4 and the Core Claim experiments.
>   * Adapter Append approaches, where learning new skills is facilitated by adapter-retrieval-based initialization [3,4], leveraging existing knowledge.
> * Additionally, implicit skill improvement and transfer can be easily achieved by appending new Gaussian components to each prototype.
> * In these cases, obsolete or infrequently used skills can be managed similarly to a file system, using either soft deletion (e.g., deprioritizing retrieval) or hard deletion (e.g., removing from memory).
>
> 5-3. suboptimal data
> We further continue the discussion of SIL-C’s robustness to noise and suboptimal data in `JDZP W2`.
>
> ---
> > W6. Concerns about terminology and phrasing that may cause confusion in context.
>
> We sincerely thank the reviewer for giving us the opportunity to clarify and improve the terminology.
> * To avoid confusion, we now use "distance" instead of the previously used term "score", and we clarify that a lower distance indicates better performance in the context of our method.
> * We replace "introduce" with "add" in line 182 to better reflect the process, since skills are added through training in response to changes in the environment, rather than being introduced independently.
>
> ---
> > W7. What are the core principles behind SIL-C, and how do the ablation results demonstrate their contributions beyond potential effects of hyperparameter tuning? ("As far as I am aware...")
>
> The core principle of SIL-C lies in adopting a lazy learning paradigm that modularizes prototype memory at each level of the hierarchy, enabling efficient trajectory distribution similarity matching through memorization. This follows up on our responses in  `Core Claim`, `JDZP W1`, `39nR W2`.
> * Type 1 baselines, which represent simple combinations of skill learning and continual learning, retain individual policy performance. However, since they lack modular integration for skill reuse, they fail to achieve BwSC, which is the main objective of our work.
> * Type 2 methods benefit from predefined labels for future skills, but we demonstrate that SIL-C achieves comparable performance without such foresight. As shown in the Kitchen Explicit scenario of Table 1, these methods degrade sharply when future skills are not known in advance, underscoring their limitations in realistic lifelong learning settings.
> * Therefore, the core strength of SIL-C lies in its ability to maintain past capabilities while seamlessly integrating future skills, without requiring prior knowledge of future tasks. This is made possible through its append-only modular updates and memory-based lazy learning, which together enable robust and scalable backward-compatible policy composition.
>
> To justify the ablation study in Figure 5, we will include the following clarification in the revised version:
> * The "SIL-C w/ random policy" setting is designed to evaluate how accurately decision-making can be achieved using only the skill hooking mechanism(trajectory similarity matching), since most subtask outputs from the high-level policy are not directly compatible in this case.
> * The results suggest that lazy learning alone offers minimal decision-making capability via skill matching, but still underperforms compared to using a synchronized high-level policy.
> * "SIL-C w/ learned policy" further improves performance by using out-of-distribution detection to identify and correct mismatches between the subtask and selected skills, which is crucial in long-horizon tasks where errors are costly.
>
> ---
> > Formatting concerns.
>
> We followed the official format, which allows flexible citation styles as long as they are used consistently.
>
> ---
> [1] Atkeson, Christopher G., Andrew W. Moore, and Stefan Schaal. "Locally weighted learning." Artificial intelligence review 11.1. 1997.
>
> [2] Douze, Matthijs, et al. "The faiss library." arXiv preprint. 2024.
>
> [3] Lee, Daehee, et al. "Incremental learning of retrievable skills for efficient continual task adaptation." NeurIPS. 2024.
>
> [4] Guo, Yijie, et al. "Srsa: Skill retrieval and adaptation for robotic assembly tasks." ICLR 2025.

---

> > ### Comment · Reviewer_UFGb · 2025-08-05
> > **Response to Rebuttal**
> >
> > I appreciate the author's changes to improve the clarity, and believe that it is improved with the proposed changes, and will raise my score.

---

> > > ### Author Response · Authors · 2025-08-05
> > >
> > > We sincerely appreciate the reviewer's time and effort in reviewing both our submission and our responses, and your valuable feedback has helped us improve the clarity and overall quality of our work.

---

### Official Review · Reviewer_39nR · 2025-07-03

**Clarity:** 2
**Significance:** 3
**Originality:** 3
**Rating:** 5
**Confidence:** 2

**Summary:**

In this paper, the authors introduce SIL-C, a novel framework for Skill Incremental Learning (SIL) that maintains compatibility between evolving skills and downstream hierarchical policies. That is, SIL-C addresses the challenge of ensuring that updates to low-level skills do not invalidate or require retraining of high-level policies. The proposed approach decouples policy learning from skill updates using a bilateral lazy learning interface, which dynamically maps high-level subtasks to low-level skills based on trajectory similarity. This interface allows for both forward and backward skill compatibility; that is, (1) the newly added skills will be used effectively when learning future policies, and (2) existing policies can, when applicable, use the newly added skills without re-training. The proposed method is evaluated on two different environments and compared to relevant existing techniques.

**Questions:**

1. What are possible alternatives to the lazy learning interface proposed in SIL-C? For example, could a learned (parametric) subtask-to-skill mapping perform similarly or better in practice? It would be helpful to understand the motivation for using a lazy, instance-based approach over other alternatives, and whether this choice contributes significantly to performance.
2. How sensitive is SIL-C to the quality of the unsupervised clustering used? The performance of the method seems highly dependent on the clustering results, but there is little discussion or ablation on this. How were hyperparameters (e.g., number of clusters, clustering algorithm) selected, and how robust is SIL-C to suboptimal clustering outcomes?
3. How many random seeds were used for the experiments presented in Tables 1 and 2? The reported standard deviations suggest significant variance, so knowing the number of seeds is important to assess statistical significance. Were statistical tests performed to validate the significance of SIL-C’s improvements over baselines?

**Ethical Concerns:**

["NO or VERY MINOR ethics concerns only"]

**Final Justification:**

Most of my concerns were addressed, and I believe this paper would be a nice contribution to the field. I increased my score, and I expect the authors to include some of the discussions from the rebuttal into the final version of the paper (if accepted).

**Limitations:**

Yes

**Paper Formatting Concerns:**

No major formatting issues.

**Quality:**

2

**Strengths And Weaknesses:**

The paper addresses an important and underexplored challenge in lifelong learning: how to maintain compatibility between evolving low-level skills and higher-level task policies. The ability to incrementally integrate skills without retraining existing policies is an important contribution with practical implications for robotics and hierarchical RL.

The paper is generally well-written, and the motivation is clearly laid out. Figure 2 is helpful in conveying the structure of the proposed system, but it is dense. I believe the paper could benefit from a high-level pseudocode or flow diagram that lays out the full execution and learning process (e.g., when are classifiers updated? how is skill hooking triggered?)

The architecture introduces significant complexity (e.g., bilateral lazy-learning interface, trajectory similarity computations, prototype memories, etc.), but the design choices are not clearly justified. There is very little intuition/high-level discussion and description of the elements, and there are no ablations or comparisons to simpler or parametric alternatives, making it unclear whether the full complexity is necessary or beneficial.

---

> ### Author Rebuttal · Authors · 2025-07-30
>
> We sincerely appreciate the reviewer’s time and effort in evaluating our work, particularly their recognition of the practical importance of maintaining skill-policy compatibility in lifelong learning agents, along with the constructive suggestions to improve clarity and the thoughtful remarks that highlight the significance of our contribution.
>
> ---
> > W1. Paper could benefit from a high-level pseudocode or flow diagram that lays out the full execution and learning process (e.g., classifier updates, skill hooking), and Figure 2 is dense.
>
> In the revision, we will add a high-level pseudocode before the current Algorithm in Appendix A.2 to clarify the overall execution and learning process, including the flow of inputs and outputs across modules. We will also simplify Figure 2 into a flow diagram aligned with the pseudocode to improve clarity.
>
> The pseudocode is as follows:
> ```python
> ## Initialization
> from collections import defaultdict
> from scipy.stats import chi2
>
> Interface = {
> 	"skills":   defaultdict(lambda: {"subgoal": [], "state": []}),
> 	"subtasks": defaultdict(lambda: defaultdict(list)),
> } # Gaussian component = (mean, precision)
> π_l, π_h = None, {}
>
> def label_dist(x, comps): # Eq.(5)
> 	def maha(x, comp):
> 		μ, Σ_inv = comp
> 		d = x - μ
> 		return d.T @ Σ_inv @ d
> 	return min(maha(x, c) for c in comps)
>
> def χ2_thr(comps, conf=0.99):
> 	dim = comps[0][0].shape[0]
> 	return chi2.ppf(conf, dim)
>
> ## Policy compatible skill incremental learning
> for p in range(1, P + 1):
> 	# Skill space update Eq.(6)
> 	χ_gz, χ_sz = create_skill_prototypes(cluster_skills(𝓓_p))
> 	for z in χ_gz:
> 		Interface["skills"][z]["subgoal"].extend(χ_gz[z])
> 		Interface["skills"][z]["state"].extend(χ_sz[z])
> 	# Skill incremental learning
> 	π_l = train_low_level_decoder(𝓓_p, Interface["skills"], π_l)
>
> 	# Downstream policy learning
> 	for τ in 𝓣_p:
> 		# Subtask space update Eq.(7)
> 		χ_g = create_subtask_prototypes(cluster_subtasks(𝓓_τ))
> 		for g in χ_g:
> 			Interface["subtasks"][τ][g].extend(χ_g[g])
>
> 		# Policy learning on the interface Eq.(8)
> 		labels = assign_subtask_labels(𝓓_τ, Interface["skills"], π_l)
> 		π_h[τ] = train_high_level_policy(𝓓_τ, labels)
>
> ## Policy inference via the interface
> def infer_action(s, τ, conf=0.99):
> 	# High-level policy: subtask proposal
> 	z_h = π_h[τ](s)
> 	g   = predict_subgoal(s, Interface["subtasks"][τ])
> 	δ   = χ2_thr(Interface["skills"][z_h]["subgoal"], conf)
>
> 	# Interface: skill validation and skill hooking Eq.(9)
> 	if label_dist(g, Interface["skills"][z_h]["subgoal"]) <= δ:
> 		z_l = z_h
> 	else:
> 		cand = [z for z in Interface["skills"]
> 			if label_dist(g, Interface["skills"][z]["subgoal"]) <= δ] + [z_h]
> 		z_l  = min(cand, key=lambda z:
> 			label_dist(s, Interface["skills"][z]["state"]))
>
> 	# Low-level decoder: action execution
> 	return π_l(s, z_l)
> ```
> ---
> > W2. The architecture introduces significant complexity (e.g., bilateral lazy-learning interface, trajectory similarity computations, prototype memories, etc.), but the design choices are not clearly justified.
>
> We clarify the detailed connection between the motivations and the implementation in the SIL-C design, aiming to support modularity, compatibility, and efficiency as follows:
> * **Motivations**
>   * To ensure **modularity** between pre-trained and adapted modules for long-term maintainability (e.g., in a 1:N producer-consumer configuration) in lifelong agents operating in complex decision-making scenarios (e.g., long-horizon tasks).
>   * To ensure **compatibility** and efficiency in the interaction between pre-trained and adapted modules during deployment.
>   * To explore mechanisms for improving agent performance **without retraining**, beyond merely maintaining the performance of the existing policy.
>
> * **Architecture design**
>   * **Hierarchical skill-based agent**: This structure supports both the reuse of pre-trained knowledge and the composition of new behaviors, aligning with practices in robotic learning for complex task solving.
>   * **Bilateral interface module**: This component connects high-level subtask specifications to low-level skill modules (both adapted and pre-trained), ensuring compatibility and controlled information exchange.
>   * **Lifelong learning interface**: This design supports efficient incremental adaptation in scenarios where future data distributions are not fully known, providing flexibility for unforeseen tasks.
>
> * **Implementation**
>   * **Skill representation**: Skills are represented using trajectory information from the streamed dataset, segmented into distinct skill groups via unsupervised clustering and encoded as skill prototypes.
>   * **Trajectory matching**: To match a given subtask with a suitable skill, we compute trajectory similarity through an instance-based classification process that aligns subtask prototypes with skill prototypes, determining executability via distance metrics and out-of-distribution rejection.
>   * **Instance-based classifiers for lazy learning interface**: Due to the unpredictability of future tasks and the high variability of behavior, learning a generalizable mapping function is challenging. To address this, we avoid irreversible updates and instead use a lazy learning approach with bilateral instance-based classifiers. For efficiency, each skill's trajectory is compressed into skill prototypes derived from sub-clusters (e.g., selecting subgoal states closest to the average across transitions), enabling the similarity estimation to be reformulated as instance-based classification with dynamic alignment.
>
> To address the reformulated classification task and enhance performance, we refine the instance-based classifier as follows:
> * **Prototype memories**: Each prototype comprises multiple sub-clusters to support fine-grained decision boundaries, as motivated by prior works [1,2,3]. Performance improvements are shown in Table 12 (Appendix C.3).
> * **Gaussian mixture and sub-clusters**: Prototypes are modeled as Gaussian mixtures, with retrieval based on Mahalanobis distance to capture dense and expressive distributions. This enables distribution-aware similarity, enhances interpretability, and supports out-of-distribution detection crucial for scalable matching in lifelong agents.
>
> ---
> > Q1. What are possible alternatives to the lazy learning interface proposed in SIL-C? Could a learned (parametric) subtask-to-skill mapping perform similarly or better in practice?
>
> A key requirement of a subtask-to-skill mapping for lifelong learning agents is common contract, which ensures the validity of selected subtask and supports dynamic relabeling at inference time. A simpler implementation of such an interface is a supervised-learned parametric mapping, which involves training on labeled subtask-skill pairs. However, this approach presents several limitations:
> * Parametric subtask-to-skill mapping
>   * Infeasible as ground-truth subtask-skill pairs are rarely available.
>   * Even if such labels are available, training requires coordination between high-level and low-level modules, undermining their modularity.
>   * As adding new skills typically requires aligning with subtasks, reintroducing retraining that the interface is intended to avoid.
> * Parametric classifier for SIL-C framework.
>   * Additional parametric classifiers for skill validation and hooking introduce extra complexity, particularly for class incremental learning on the skill incremental learning.
>   * Parametric knowledge is difficult to decouple once learned, which hinders maintainability and efficiency in tasks like creating, updating, and deleting skills over time.
>
> ---
> > Q2. Provide discussions of dependency of SIL-C on unsupervised clustering quality and analysis and hyperparameter selection justifications.
>
> As shown in Figure 8 (Appendix C.2), we observe that performance generally improves as the number of clusters increases, particularly during both the SIL and policy learning stages. This suggests that finer clustering leads to a more diverse and expressive skill set, contributing to improved policy learning and task completion.
>
> Our hyperparameter selection process followed principled steps based on both prior literature and empirical findings:
> * We initially adopted skill clustering setting from the PTGM [4] baseline (Type 1), which uses 50 clusters for entire dataset. For phase-wise skill learning, we divided these into 4 temporal segments, yielding 12 skills per phase.
> * However, this configuration produced suboptimal results. Based on observations from prior work [4] that performance improves with larger cluster sizes, we empirically increased the number of clusters to 20 skills per phase.
> * For fair comparison, SIL-C adopted the same underlying clustering configuration and algorithmic hyperparameters with Type 1 baselines.
>
> ---
> > Q3. How many seeds were used for Tables 1-2, and were statistical tests conducted to confirm SIL-C's improvements?
>
> We used four random seeds for the experiments presented in Tables 1 and 2, and this detail will be added to each table caption for clarity.
>
> We performed p-value tests to validate SIL-C's gains.
> * SIL-C matches or outperforms baselines without using skill labels, even in settings where future skills are unknown.
> * Emergent SIL scenario in Table 1 shows statistically significant improvements over all baselines except those using explicit labels, where the difference is less significant (p > 0.05).
>
> [1] Wistuba, Martin, et al. "Continual learning with low rank adaptation." arXiv preprint. 2023.
>
> [2] Lee, Daehee, et al. "Incremental learning of retrievable skills for efficient continual task adaptation." NeurIPS. 2024.
>
> [3] Wang, Yanjie, et al. "High-dimension Prototype is a Better Incremental Object Detection Learner." ICLR. 2025.
>
> [4] Yuan, Haoqi, et al. "Pre-training goal-based models for sample-efficient reinforcement learning." ICLR. 2024.

---

> > ### Comment · Reviewer_39nR · 2025-08-06
> >
> > I appreciate the authors' detailed response, it clarifies many of my questions. Based on this, I will raise my score. I do urge the authors to incorporate these discussions into a final version of the paper.

---

> > > ### Author Response · Authors · 2025-08-07
> > >
> > > We sincerely appreciate the reviewer’s time and effort in reviewing both our submission and our responses. We have reflected the discussions with the reviewer in the corresponding sections of the manuscript and ensure that they will also be included in the final version.

---

### Official Review · Reviewer_PQiK · 2025-07-03

**Clarity:** 3
**Significance:** 3
**Originality:** 3
**Rating:** 5
**Confidence:** 3

**Summary:**

The paper presents a method of Skill Incremental Learning by introducing a lazy learning interface that connects subtasks to skills and a policy integration mechanism that enables seamless integration of learned skills to existing downstream policies. The key problem that this method tries to tackle is the forgetting of past skills as well as the usefulness of newly added skills for downstream tasks. New skills at different phases is determined using clustering. The paper evaluates this method on both backwards and forwards compatibility; it outperforms the baselines in overall performance and the forward compatibility, and performs closely to methods with predefined skill labels.

**Questions:**

- Line 95: In what cases would $\psi$ not be the identity map and why?
- Line 99-100: Would $\mathcal{T}_p$ depend on the dataset $\mathcal{D}_p$ that is presented to the agent? I.e., would the set of evaluation tasks be setup to best use the dataset, or would it be based on some other criterion (if any)?
- Line 101 "... assume access to task-level expert demonstrations..." -> Does this mean that the agent gets examples of how to incorporate the newly acquired skills, rather than learning whether they are useful or not itself?
- Line 125: Is there a difference between a "subgoal" and a "subtask" in the context of the paper?

**Ethical Concerns:**

["NO or VERY MINOR ethics concerns only"]

**Final Justification:**

I am happy with maintaining my score provided that some of the responses to my questions are discussed in later revisions of the paper (if it proceeds). I believe that this is a strong, interesting paper.

**Limitations:**

Yes

**Quality:**

4

**Strengths And Weaknesses:**

## Strengths
- Generally well-written and with reasonably clearly written motivations and method. See minor comments below for improvements with the exposition.
- Good literature review.
- The method is an interesting idea.
- A good range of experiments were carried out with a good number of baselines.
- Good ablation study on removing the lazy learning interface.
## Weaknesses
- I think that the paper could use another small iteration where some improvements to clarity are made (see suggestions below and Questions).
#### Typos and suggestions
- Line 27: "Yet, a critical challenge remain**s**..." (remain -> remains).
- Line 95: "... may be the identity..." -> "... may be the identity **map** ..."; I think adding map makes it clearer.
- Line 100: Please define what $\tau$ is right next to $\mathcal{D}\_\tau$ rather in line 102. E.g., "Let $\tau$ denote a task; we assume access to task-level expert demonstrations $\mathcal{D}_\tau$ ...".

- Line 103: Could you add "Our objective is to find the collections of high- and low-level optimal parameters over all phases, $(\Theta\_h\^\*, \Theta\_l\^\*)$ ...". This is so that it immediately clear what $(\Theta_h^\*, \Theta_l^\*)$ are without having to wait till the end of Equation 2.

- Section 4 paragraph 1: Could the appropriate mathematical notation for terms such as "subtask space", "skill space", "task space", etc.
	- The same in Section 4.2 paragraph 1.
- Section 5: Can the number of repetitions be added to the captions of tables with $\pm$ values.

---

> ### Author Rebuttal · Authors · 2025-07-31
>
> We thank the reviewer for the clear summary and for recognizing the main contributions of our work. We also appreciate the positive comments on our method, writing, and experiments.
> In addition, we are grateful for the questions, which reflect a solid understanding of our approach and raise important points about its broader applicability. These help us clarify the motivation and design choices behind our method.
>
> ---
> > W1.  Clarity improvements.
>
> We will incorporate the proposed suggestions across Sections 1 to 5 to improve clarity and precision throughout the paper.
>
> * Specifically, we will clarify the notation for "subtask space" and "skill space" as $\mathcal{X}\_{h}$ and $\mathcal{X}\_{l}$, respectively.
> * In addition, we will revise the term "task space" in line 112 to explicitly denote *subtask space*, to avoid ambiguity.
> * Regarding the reviewer's suggestion on table captions with $\pm$ values, we confirm that the reported values reflect the mean and standard deviation across 4 different random seeds (repetitions). We will revise the main table captions to explicitly indicate this for clarity, with this information currently stated in Appendix B.5.
>
> ---
> > Q1. In what cases would $\psi$ not be the identity map and why?
>
> The mapping function $\psi$ is introduced to reconcile the representational discrepancy between the high-level policy, which operates over a subtask and the low-level skill decoder, which consumes inputs from a skill.
>
> Although $\psi$ reduces to the identity function when both modules are co-designed to share an ideally aligned latent space, such alignment is not guaranteed in practical settings. This is especially true in lifelong learning scenarios, where policies must continually adapt to evolving conditions. Even in co-designed systems, factors such as distributional shifts, task-specific embedding variations, or implementation-level inconsistencies can still require a non-trivial mapping.
>
> The need for a non-identity $\psi$ becomes particularly evident when the high-level and low-level modules are trained asynchronously or with different algorithms. In such cases, their internal vocabularies may drift apart. Two common situations where this occurs are:
> * In related works such as PTGM, the high-level and low-level policies are trained independently, often using distinct objectives and datasets. The high-level policy typically outputs a symbolic subtask (e.g., a behavior embedding), while the low-level policy expects a subgoal state as input. A hash-based lookup function $\psi$ is commonly used to bridge this semantic gap.
> * In lifelong learning scenarios, the alignment between the high-level output space and the low-level input space may shift over time. These shifts can occur even when task semantics are preserved, as a result of ongoing learning or environmental changes. In such cases, assuming $\psi$ is the identity function becomes invalid, and updating the mapping is necessary to maintain effective coordination between components.
>
> ---
> > Q2. Would $\mathcal{T}_p$ depend on the dataset $\mathcal{D}_p$ that is presented to the agent?
> I.e., would the set of evaluation tasks be setup to best use the dataset, or would it be based on some other criterion (if any)?
>
> For, problem formulation, the set of evaluation tasks $\mathcal{T}_p$ is independent of the dataset $\mathcal{D}_p$ used for skill incremental learning. Most incremental learning settings assume that future tasks are unknown in advance, and our problem formulation reflects this assumption.
> * For skill-policy compatibility evaluation, as noted in Appendix B.1.2, in our experiments we selected 24 tasks from the base tasks included in the Franka-Kitchen mixed dataset. This setup assumes that all skills have been sufficiently learned by the final phase of SIL, such that successful execution of these tasks is reasonably expected. This design choice is intended to evaluate skill-policy compatibility , under the assumption that learned skills can be appropriately applied to the evaluation tasks.
> * Including unseen tasks or tasks that are inherently more difficult to learn could introduce confounding factors, making it unclear whether performance degradation is due to limitations in skill-policy compatibility or simply due to task difficulty.
> * Therefore, structuring the evaluation tasks based on the dataset is a deliberate choice to ensure clarity in evaluation results.
>
> As a possible real-world use case that can arise in the producer-consumer deployment configuration for these hierarchical agents:
> * Here, the producer provides a task-agnostic pre-trained model, i.e., trains the skills, while the consumer fine-tunes a modular policy to make it task-specific.
> * In such scenarios, building on our evaluation protocols for FwSC and BwSC, the construction of $\mathcal{T}_p$ could incorporate objectives such as maximizing consumer rewards in task composition.
> * Likewise, clearly formulating the problem of skill-policy compatibility and designing $\mathcal{T}_p$ accordingly is essential for ensuring a robust and meaningful evaluation.
>
> In practical skill incremental learning scenarios, factors like domain shift or noisy skill distributions may arise. Extending our evaluation framework to assess generalization under such conditions, such as evaluating performance on tasks with perturbed state-action transitions, would be valuable. We provide an example of such experimental scenarios in our response to `JDZP Q2`, which illustrates the model’s performance under these challenging conditions.
>
> ---
> > Q3. "... assume access to task-level expert demonstrations..." -> Does this mean that the agent gets examples of how to incorporate the newly acquired skills, rather than learning whether they are useful or not itself?
>
> Yes. In our work, we assume that the agent learns from training data that contains objective about the task which to be accomplished. The provided expert demonstrations implicitly contain examples of how to compose the relevant skills, either in a dense or sparse manner. Importantly, these demonstrations are not used directly during evaluation or after the policy and subtask prototypes have been learned.
>
> To relax and generalize this assumption, we designed the few-shot imitation learning experiments in Table 2. Specifically, in addition to full demonstration trajectories, we include sparse examples obtained by downsampling the trajectories to partial versions (e.g., 50% and 20% of the original trajectory). These partial trajectories are often insufficient for full behavior cloning. Despite the limited information, the results in Table 2 clearly show that SIL-C performs effectively under these constrained conditions.
>
> To address the reviewer's insightful question regarding the generality of our framework, we provide an extended discussion in the Future Directions section. Based on our current formulation, we believe several meaningful extensions are possible, opening up exciting avenues for future research:
> * **Minimal task supervision**: SIL-C could be extended to more weakly supervised settings, potentially even to scenarios where only a high-level goal is provided. In such cases, the agent could learn to chain prototypes within the interface to infer plausible skill execution paths without explicit demonstrations.
> * **Skill usage analysis for practical deployment**: SIL-C could also be leveraged as an analysis tool in large-scale deployment scenarios. By tracking how often and in what contexts certain skills are referenced, it could serve as a guide for data collection or for continual adaptation of skill libraries.
> * **Safety-oriented skill validation**: The Gaussian prototype structure also allows for potential safety extensions. For example, it may help prevent the execution of unsafe or unvalidated skills in real-world environments by identifying anomalies in prototype activations.
> * **Reward-based skill hooking without task demonstrations**: Similar to other skill-based learning methods, reinforcement learning acceleration can be indirectly observed through improved FwSC. However, as discussed earlier, achieving BwSC generally requires access to task-level information, since the agent must know what it is trying to improve. That said, it would be interesting to explore reward-driven skill hooking, where the agent hooks into relevant skills purely from learning signals, even without any demonstration data. This direction may prove valuable for developing more autonomous lifelong embodied agents.
>
> Further clarifications on the motivation behind our architectural design and its implementation can be found in our responses to the following questions:
> 1. Motivation for Architecture Design - `JDZP W1 Q1`
> 2. Connection between motivations and implementation - `39nR W2`
>
> ---
> > Q4. Is there a difference between a "subgoal" and a "subtask" in the context of the paper?
>
> Yes, there is a difference. A subtask refers to an abstract unit of behavior selected by the policy, whereas a subgoal is a concrete intermediate state that the agent aims to reach in order to carry out that subtask.

---

> > ### Comment · Reviewer_PQiK · 2025-08-06
> > **Response to Rebuttal**
> >
> > Thank you very much for your detailed response! I think my concerns are addressed! Hopefully some of the responses to the questions I asked could make their way into the manuscript to be clearer to other readers who might have similar questions.

---

> > > ### Author Response · Authors · 2025-08-07
> > >
> > > We sincerely appreciate the reviewer's time and effort in reviewing both our submission and our responses! We have carefully incorporated the responses to the reviewer’s questions into the manuscript to ensure that readers with similar questions can clearly understand our work. Thank you again!

---

### Author Response · Authors · 2025-08-04

Thank you for your time and thoughtful effort in reviewing our paper. As the discussion deadline approaches, we kindly ask the reviewer to take a moment to review our response and share any additional comments or questions.

Summary of our responses:
* **`PQiK`**
  * Improved presentation and clarifications
  * Additional discussion on the extended applicability of SIL-C
* **`39nR`**
  * Improved presentation and clarifications (with pseudocode)
  * Clarification on the design choices in SIL-C’s implementation and their underlying motivations
  * Discussion on the applicability of parametric alternatives
  * Discussion and rationale for hyperparameter selection
* **`UFGb`**
  * Clarification of the core claim and inclusion of a motivating experiment
  * Improved presentation and clarifications
  * Discussion on the scalability of the framework
  * Analysis of ablation study impact
* **`JDZP`**
  * Clarification of the motivation behind the architectural setup
  * Robustness analysis of SIL-C to noisy inputs

---

### Note · Authors · 2025-08-11

We close the Author-Reviewer discussion period with a summary of the reviews and discussions, and sincerely thank the NeurIPS community for their time and constructive engagement.

---
> ### **Strengths**

* Clearly addresses the practical and important challenge of maintaining *skill-policy compatibility* in lifelong learning (*Skill Incremental Learning*, SIL), with an explicit formulation of *Forward Skill Compatibility (FwSC)* and *Backward Skill Compatibility (BwSC)*. `PQiK` `39nR` `UFGb` `JDZP`
* Clearly states the motivation. `PQiK` `39nR` `JDZP`
* Presents an interesting idea for a *bilateral lazy learning interface* with instance-based classifiers that perform trajectory similarity matching for *subtask-to-skill mapping*. `PQiK` `UFGb` `JDZP`
* Includes extensive experiments in *challenging, long-horizon environments* under various SIL scenarios. `PQiK` `UFGb` `JDZP`

> ### **Raised Concerns and Our Responses**

* **Presentation & clarity** `PQiK` `39nR` `UFGb`
  * We streamlined notation, refined terminology and expressions, simplified dense figures, added high-level pseudocode, and improved overall readability.
* **Architectural motivations and design choices** `39nR` `UFGb` `JDZP`
  * We expanded the rationale for adopting a lazy, instance-based interface within a hierarchical architecture, explicitly linking these choices to our goals of modularity, compatibility, and efficient skill reuse without retraining.

> ### **Discussion Points**
* **Broader applicability** of SIL-C under the assumptions in our problem formulation. `PQiK`
* **Feasibility** of *parametric alternatives* for subtask-to-skill mappings versus our *lazy learning* approach. `39nR`
* **Scalability and long-term maintainability** of the *prototype memory* mechanism. `UFGb`
* **Robustness** of SIL-C to *observation noise*. `JDZP`

We appreciate the constructive feedback, which has helped improve the clarity and technical foundation of our work. The final version will incorporate these insights to present a more precise, scalable, and broadly applicable **SIL-C** framework.

---

### Decision · Program_Chairs · 2025-09-17

**Decision:**

Accept (spotlight)

**Comment:**

The authors propose a novel framework for Skill Incremental Learning (SIL) that maintains compatibility between evolving skills and downstream hierarchical policies. It addresses the challenge of ensuring that updates to low-level skills do not invalidate or require retraining of high-level policies. The proposed approach decouples policy learning from skill updates using a bilateral lazy learning interface, which dynamically maps high-level subtasks to low-level skills based on trajectory similarity. The proposed method outperforms the baselines in overall performance and forward compatibility, and performs closely to methods with predefined skill labels.
Most concerns are addressed through clarification and additional experiments. The final recommendations are two acceptances and two borderline acceptances. One reviewer would like to see more real-world experiments. However, the reviewer still gave borderline acceptance. Hence, the AC recommends acceptance.